# Pan evaporation is increased by submerged macrophytes

Brigitta Simon-Gáspár[1], Gábor Soós[1], Angela Anda[1]

[1]Institute of Agronomy, Georgikon Campus, Hungarian University of Agriculture and Life Sciences, 8360 Keszthely, Hungary

*Correspondence to*: Brigitta Simon-Gáspár (simon.gaspar.brigitta@uni-mate.hu)

5 **Abstract.** The topic of evaporation estimates is fundamental to land-surface hydrology. In this study, FAO-56 Penman-Monteith equation (FAO56-PM), multiple stepwise regression (MLR) and Kohonen self-organizing map (K-SOM) techniques were used for the estimation of daily pan evaporation ($E_p$) in three treatments, where C was the standard class A pan with top water, S was a pan with sediment covered bottom, and SM was class A pan containing submerged macrophytes (*Myriophyllum spicatum*, *Potamogeton perfoliatus*, and *Najas marina*), at Keszthely, Hungary, in a six-season experiment, between 2015 and 10 2020. The modelling approach included six measured meteorological variables. Average $E_p$ varied from 0.6 to 6.9 mm d$^{-1}$ for C, 0.7 to 7.9 mm d$^{-1}$ for S, whereas from 0.9 to 8.2 mm d$^{-1}$ for SM during the growing seasons studied. Correlation analysis and K-SOM visual representation revealed that air temperature and global radiation had positive correlation, while relative humidity had a negative correlation with the $E_p$ of C, S and SM. The results showed that the MLR method provided close compliance ($R^2$=0.58-0.62) with the observed pan evaporation values, but the K-SOM method ($R^2$=0.97-0.98) yielded by far 15 the closest match to observed evaporation estimates for all three pans.

To our best knowledge, no similar work has been published previously using the three modelling methods for seeded pan evaporation estimation.

The current study differs from previous evaporation estimates by using neural networks even with those pans containing sediments and submerged macrophytes. Their evaporation will be treated directly by K-SOM, in which the modelling is more 20 than the simple Ep of a Class A pan filled with clean tap water.

## 1 Introduction

Open water evaporation is one of the paramount elements of the hydrological cycle (Brutsaert, 1982). Evaporation losses from various surfaces appear to be increasing in recent decades (Mbangiwa et al., 2019). Due to climate change, it is also extremely important to determine evaporation as accurately as possible (Fournier et al., 2021), for which both direct and indirect methods 25 are available. As a direct method, the evaporation pans (primarily the class A pan proposed by the World Meteorological Organization, WMO are used extensively throughout the world to measure open water evaporation and to estimate reference evapotranspiration (Rahimikhoob, 2009; Fuentes et al., 2020). Measurements of pan evaporation may be spatially and temporally limited (Jensen et el., 1990; Rahimikhoob, 2009), like in case of maintenance problems which can affect the

accuracy of evaporation measurements, e.g., most often turbidity of water, watering of birds or other animals (Tabari et al.,
30  2010).

To indirectly determine evaporation, several methods can be used: empirical equations are applied that estimate evaporation based on meteorological variables (air temperature, $T_a$, relative humidity, RH, global radiation, $R_s$), or transfer and water budget methods (Burman, 1976). The most widely used empirical formula is a FAO-56 Penman–Monteith equation (FAO56-PM) (Allen et al., 1998), which is the standard method for computation of daily reference evapotranspiration. However,
measuring meteorological variables requires sophisticated instruments, which can often be challenging (Arunkumar and Jothiprakash, 2013; Sattari et al., 2020). The amount of required data and the difficulty of the estimation of the unknown meteorological elements may be additional problems (Sanikhani et al., 2015; Khatibi et al., 2020). Therefore, there is a need for alternative methods that are simple and effective, require fewer inputs and are also able to solve problems which are difficult to formalize (Sudheer et al., 2003; Kisi, 2015; Malik et al. 2020a).

A promising tool that can be used to estimate $E_p$ and is a suitable alternative to the empirical models is the different neural networks (Kim et al., 2015), thus the neural networks are increasingly used in evaporation and evapotranspiration estimation (Kumar et al., 2002; Keskin and Terzi 2006; Rahimikhoob, 2009; Alsumaiei, 2020). The machine learning techniques can map high dimensional data to a low dimensional space and show some similar properties based on internal data relationships (Pearce et al., 2011; Zelazny et al., 2011). In recent years, machine learning techniques have been broadly employed in hydrological
and environmental models, including to forecast evaporation (Wu et al., 2020). Numerous results in the literature indicate that machine learning algorithms such as artificial neural network (ANN), M5 model tree (M5T), support vector machines (SVM), multivariate adaptive regression splines (MARS), gradient boosting with categorical features support (CatBoost), random forest (RF) perform excellently in predicting pan evaporation as well (Dong et al., 2021).

Of the available methods, Self Organizing Maps (SOM) is able to handle noisy, irregular and multivariate data well (Nakagawa,
2017). As a result, it has become one of the most popular neural network (NN) methods for data analysis (Nada et al., 2017). SOMs are used in many disciplines (Nakagawa 2017, 2020), such as agriculture (Li et al., 2019; Kumar et al., 2021a,b), ecology (Bedoya et al. 2009, Ristić et al., 2020), hydrology (Guntu et al., 2020; Rivas-Tabares, 2020; Lee and Kim, 2021), meteorology (Nada et al., 2017; Berkovic et al., 2021, Doan et al. 2021), and water management (Gu et al. 2019, Gholami et al., 2020; Lee et al., 2021). The unsupervised NNs, including Kohonen Self Organizing Maps (K-SOM), have several
advantages (Kohonen, 1982; Kohonen, 2001). The essence of this method is to group the large-dimensional array of the input layer into a 2-dimensional array in the output layer, so that all variables of the input vectors can be found in each node of the output layer (Adeloye et al., 2011). Another advantage of K-SOM over traditional models is that it also has visualization abilities (Hadjisolomou et al., 2018).

The study site, Lake Balaton, is the largest shallow freshwater lake in Central Europe with a surface area of 596 km$^2$ (Figure
1). The three most dominant submerged macrophytes in the Lake Balaton are *Potamogeton perfoliatus*, *Myriophyllum spicatum* and *Najas marina*, therefore it was appropriate to include these three species in the observation. In Hungary, submerged macrophytes colonize in lakes in the summer season (from June to September). Evaporation of open water surfaces

is usually measured by means of pans endowed with unrealistic properties. These pans are filled with clean tap water and the evaporated water is also replaced with tap water unlike in natural ecosystems. In nature there may also be submerged

macrophytes living in the open water. The presence of these plants is essential and affects the chemical and physical water properties including its quality (Kimmel and Groeger, 1984; Zhang et al., 2017; Yan et al., 2019). Furthermore, the species that are rooted in the sediment can stabilize the sediment by inhibiting its resuspension (Madsen and Cedergreen, 2002; Vymazal, 2013).

Changes in the heat regime of a water body had been reported to result in alterations of macrophyte community composition

(Barko et al., 1982; Poikane et al., 2015; Fritz et al., 2017; Kim and Nishihiro, 2020), which may affect the temporal appearance and spatial distribution of macrophytes in the future. As a result, due to global climate change, it is important to examine submerged macrophytes in all aspects, including their effect on evaporation.

The aim of the study was to investigate the effect of littoral sediment and macrophytes on lake evaporation and not an introduction of a new method in pan evaporation estimation. The previous results in FAO-56 Penman-Monteith equation (Allen

et al., 1998), Kohonen self-organizing map techniques (Kohonen, 1982) and multiple stepwise regression are classic methods, highlighted widely by citations in the study. They are the tools in analysing the effect of sediment and macrophytes in pan (lake) evaporation estimation only. The novelty of the paper is in the way the evaporation estimation is carried out.

To our best knowledge, there are no studies attempting to project water bodies' evaporation using traditional A pan measurements, taking the macrophytes- and sediment-related factors into account under such climate conditions as our

experimental site.

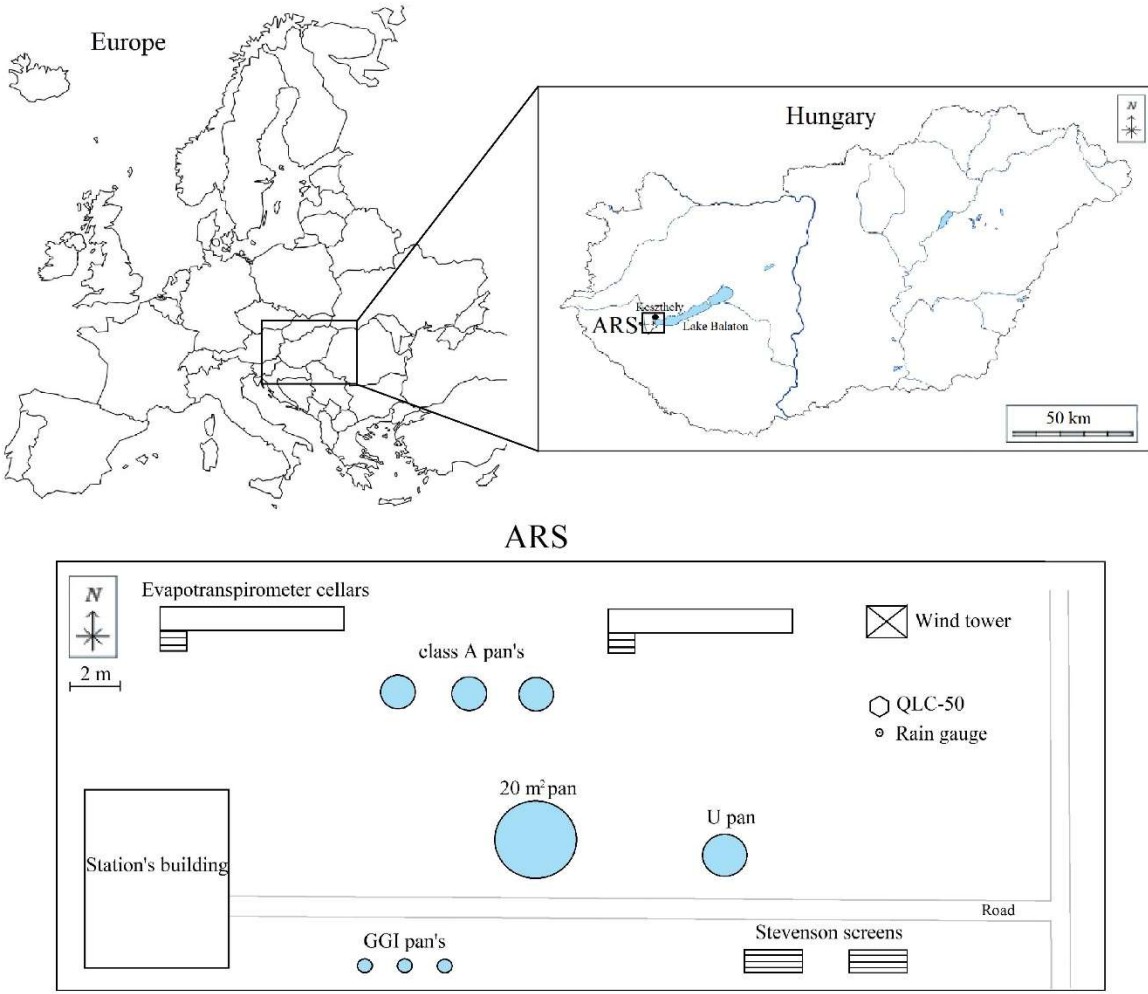

**Figure 1: Location map of the study area with Agrometeorological Research Station (ARS) at Keszthely, Hungary (from https://www.vectorstock.com).**

## 2 Materials and methods

### 2.1 Case study and data description

The climate of the region – see also Fig. 1 – is mild continental (Cfb) with warm, dry summers and fairly cold winters according to the Köppen-Geiger classification (Kottek, Grieser, Beck, Rudolf and Rubel, 2006). Months were included in the study (from June to September). Meteorological variables were recorded by a QLC-50 climate station (Vaisala, Helsinki, Finland) fitted with a CM-3 pyranometer (Kipp & Zonen Corp., Delft, the Netherlands) located at Keszthely Agrometeorological Research Station (ARS) (latitude: 46°44′N, longitude: 17°14′E, elevation: 124 m a.s.l.) between 2015-2020. The ARS is placed on the area of the Hungarian University of Agriculture and Life Sciences. With the exception of wind speed, meteorological data of

$T_a$, RH, $R_s$, daily maximum temperature ($T_{max}$), daily minimum temperature ($T_{min}$) and precipitation (P) were measured at 2 m above the ground surface. The height of windspeed (u) measurements was 10.5 m. The daily mean values of meteorological variables were calculated as average of 10 minutes observations of a 24-hour period.

In this study, class A evaporation pans were used to determine daily evaporation ($E_p$). The class A pans were 1.21 m in diameter and 0.25 m in height located on an elevated (~0.15 m) wooden grid, with a water surface area of ~1.15 $m^2$. The daily rate of $E_p$ was calculated from the difference in water level for two consecutive days, considering any precipitation that may have fallen into the pans. The daily water loss was measured every morning at 7.00 am LMT.

In the ARS area 3 class A pans were placed, 5 meters apart (Figure 2). A class A pan was recommended by the WMO to be
used as a standard treatment (control, C). Two class A pans were covered on the bottom with sediment to a thickness of 0.002 m (S). The used sediment was psammal/psammopelal (Ø > 6 µm - 2 mm, sand/sand with mud) with the following composition: quartz, calcite, aragonite, dolomite, muscovite, chlorite, feldspar, smectite, kaolinite and pyrite (Anda et al., 2016). Submerged, freshwater aquatic macrophytes were planted in third class A pans with sediment-covered bottom (Anda et al., 2016; Anda et al., 2018). Macrophyte samples were gathered from lake Balaton (Keszthely Bay) with similar water depth (0.6–0.8 m) each
year. The amount of crop density was controlled monthly without variation in the green mass weight of crops between natural habitat and ''seeded" Class A pans. In the experimental area three species of submerged, freshwater aquatic macrophytes: *Potamogeton perfoliatus*, *Myriophyllum spicatum* and *Najas marina* were colonized. Due to the development of submersed macrophytes, Class A pans were operational from June to September in the growing season 2015-2020.

In the last vegetation period, to detect vertical water temperature ($T_w$) profiles, four fastened thermistors of Delta Ohm HD-
226-1 (accuracy: 0.3ºC) collected the temperature data at 0.05, 0.10, 0.15 m depth from the pan bottom and on the water surface, at 10-min intervals. Hourly averaged $T_w$ values were used in the analysis. To present diurnal variation in Tw and stratification, sample days were selected for clear-sky, calm, and cloudy weather conditions.

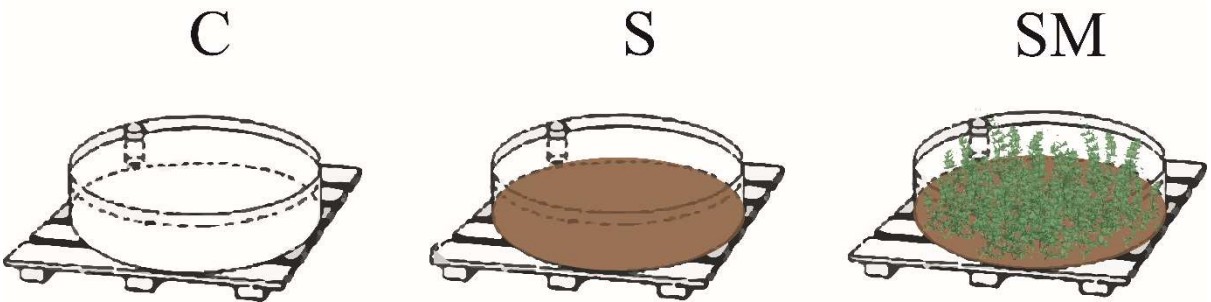

**Figure 2: Class A pans with different treatments: C, S and SM denote ''empty", sediment-covered and macrophyte-planted class A pans in the middle of the meteorological garden.**

The weather of the studied growing seasons was specified by the monthly Thornthwaite Index (TI) of the World Meteorological Organization (WMO) Report (1975):

$\quad TI = 1.65(\frac{Pm}{Tam} + 12.2)^{10/9})$ $\hspace{8cm}$ (1)

where $P_m$ and $T_{am}$ are the monthly sum of precipitation and the monthly mean air temperature, respectively.

In classifying the weather in each season's months, a 20% deviation was assumed from climate norms (1981–2010), above and below the $TI_{norm}$ for both included meteorological variables ($P_m$ and $T_{am}$), allowing the following weather classes to be distinguished:

$\quad$ Warm-dry month (h): $TI_{month} > TI_{norm}$ x 0.8;

Cooler-wet month (c): $TI_{month} > TI_{norm}$ x 1.2;

Month with normal weather (n): $TI_{norm}$ x 0.8 $\leq TI_{month} \leq TI_{norm}$ x 1.2.

By counting the highest number of months within each of these three groups, the season was considered to be either normal, cool or warm.

## 2.2 Multiple stepwise regression (MLR)

The regression models are important tools for investigating relations between dependent and independent data (Razi and Athappilly, 2005), which method has been used for a long time in the investigation of meteorological variables. Evaporation can be modelled by multiple linear regressions using different meteorological variables (e.g., Ta, RH, u) (Almedeij, 2012). The MLR can be expressed by the following equation:

$\quad y = b_0 + b_1x_1 + \cdots + b_kx_k + a$, $\hspace{7cm}$ (2)

where $b_0$, $b_1$ … and $b_k$ are fitting constant, $x_1$… and $x_k$ represent the observed meteorological variables and a is a random error term. The a is remaining effects on estimated $E_p$ (y) of variables not explicitly included in the model (Patle et al., 2020). The dependent variable, y was $E_p$.

## 2.3 FAO-56 Penman-Monteith (FAO56-PM) method

The Penman-Monteith model is considered as the international standard for computing potential evapotranspiration and predicting crop water requirement. FAO56-PM may also be proper method to get pan evaporation with submerged macrophytes. Wang et al. (2021) reported that actual evaporation is important for hydrological research due to its direct impact on the hydrologic processes (water cycle, water resources management). The above authors concluded that to estimate pan evaporation, it is essential to find the proper formulation of the Penman-Monteith equation, a special case of the multiple stepwise regression methods. It may be especially true even in pans with seeded macrophytes. In accordance with composition of lake ecosystems, this is the method in evaporation estimation that implies living organism.

The reference evapotranspiration $ET_0$ was estimated by the WMO standardized FAO-56 Penman-Monteith method (Allen et al., 1998; Allen et al., 2005) at a daily step for short reference crops (clipped grass of 12 cm) as followed:

$$ET_0 = \frac{0.408\Delta(R_n-G)+\gamma\frac{900}{T_a+273}u(e_s-e_a)}{\Delta+\gamma(1+0.34u)},$$ (3)

where $R_n$ is net radiation [MJ m$^{-2}$ d$^{-1}$], G is the soil heat flux density [MJ m$^{-2}$ d$^{-1}$], $T_a$ is the mean daily air temperature at 2 m height [°C], u is wind speed [m s$^{-1}$] at 2 m height, $e_s$ is the saturation vapor pressure [kPa], $e_a$ is the actual vapor pressure [kPa], $\Delta$ is the slope of the vapor pressure curve [kPa °C$^{-1}$], $\gamma$ is a psychrometric constant [kPa °C$^{-1}$], and 0.408 is a conversion factor from MJ m$^{-2}$ d$^{-1}$ to equivalent evaporation in mm d$^{-1}$.

$R_n$ was estimated from global radiation, mean daily temperature, the mean daily vapor pressure, the site latitude and elevation

after Allen et al. (2005). A fixed value of 0.23 was applied for the albedo. It was assumed that soil heat flux density was G= 0 on a daily basis. Detailed description of the process can be read in Soós and Anda (2014).

The Tetens equation (Monteith and Unsworth, 2008; Allen et al., 1998; Tetens, 1930) was used for calculating saturation vapour pressure ($e_s$) as follows:

$$e_s = 0.6108 * \exp\left(17.27 T_a / (T_a + 237.3)\right)$$ (4)

where $T_a$ is the air temperature in °C. The actual vapour pressure, $e_a$ was calculated from the relative humidity (RH):

$$e_a = \left(\frac{RH}{100}\right) * e_s$$ (5)

### 2.4 Kohonen self-organization map (K-SOM)

The K-SOM is a nonlinear mapping technique, which identifies groups of similarity in data sets without normal distribution

assumption (Kohonen, 1982). SOM is a powerful and effective tool for complex data analyses such as data mining, estimation, and prediction. Using SOM, informative reference vectors are obtained via iterative updates under three main successive procedures: competition with nodes (1), selection of a winner node (2) and updating of the reference vector (3) (Yu et al., 2018). Every node has its vector adjusted according to sequential algorithm with the Gaussian neighbourhood function. The SOM consists of an input layer and an output layer (Park et al., 2006), where the output layer consists of so-called neurons,

which are usually located in a hexagonal grid and are fully interconnected (Peeters et al., 2007). A schematic illustration of K-SOM is presented in Figure 3. As similar input patterns could have different outputs, to determine the best output for a given input pattern is to use the mean output value as the clustered input patterns to the correspondent neuron, and then the closest (most similar) neuron would be directly used for the given input pattern (Chang et al., 2010, Kohonen, 1990).

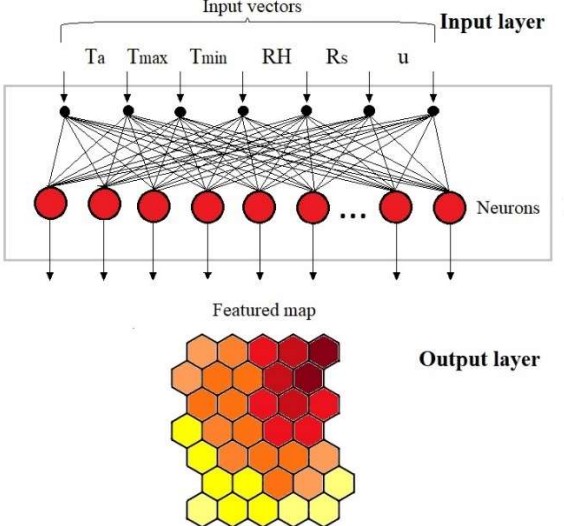

**Figure 3: Illustration of the winning node and its neighbourhood in the Kohonen self-organizing map (K-SOM).**

The importance of K-SOM in the field of environmental science lies in the fact that SOMs can be used for prediction and correlation analysis, mostly with visual representation (Barreto and Pérez-Uribe, 2007). An outstanding element of this is that K-SOM finds statistically significant dependencies among the variables in a multidimensional data sample. In the case where two variables are highly correlated, K-SOM produces two similar component planes (Barreto and Pérez-Uribe, 2007).

K-SOM as NN provides a method above the standard estimations of pan evaporation, which seems necessary to get evaporation of natural ecosystems including lakes. In other words, applying a method, in which the pan evaporation is estimated from other, easily measurable meteorological parameters such as sun radiation, air temperature and relative humidity has primary importance. This approach has widely been used for pan evaporation projection among others by Kisi et al. (2016) and Lin et al. (2013). Kisi et al (2016) compared the soft computing model K-SOM and multiple linear

regression (MLR). The authors demonstrated the superiority of K-SOM over MLR even in the model performance.

## 2.5 Statistics and performance evaluation criteria

The Shapiro–Wilk test was used as a statistical test for normality, with a chosen alpha value of 0.05 ($p<0.05$). Two-way analysis of variance (ANOVA) with Tukey's HSD test was performed to examine the impacts of treatments C, S and SM on class A pan $E_p$. To study the impact of meteorological elements on $E_p$ of C, S and SM treatments, Pearson's correlation analysis

was used. This, as well as the MLR, was carried out with SPSS Statistics software. In this study, the K-SOM algorithm was executed using MATLAB 2019b software. To train (years: 2015–2017) and test the models (years: 2018–2020), half of the data were used.

Performance of the proposed models is evaluated by computing statistical indices, such as root mean square error (RMSE), mean absolute error (MAE), scatter index (SI) and Nash-Sutcliffe efficiency (NSE) between observed and estimated values of

E_p for the data sets considered. The RMSE range is zero to infinity ($0 < RMSE < \infty$); the lower the RMSE, the better the model's performance. The RMSE is proportional to the observed mean, as a result SI (Shiri and Kişi, 2011) forms a good non-dimensional error measure. NSE (Nash and Sutcliffe 1970; ASCE 1993) compares the congruence between the observed and predicted data. A high value of NSE ($NSE \leq 1$) indicates high efficiency of the model (Duan et al., 2016; Li and Liu, 2020). These evaluation criteria calculate as following equations:

$$RMSE = \sqrt{\frac{\sum_{i=1}^{n}(E_{p_{obs,i}} - E_{p_{est,i}})^2}{n}}, \tag{6}$$

$$MAE = \frac{\sum_{i=1}^{n}|E_{p_{est,i}} - E_{p_{obs,i}}|}{n}, \tag{7}$$

$$NSE = 1 - \frac{\sum_{i=1}^{n}(E_{p_{obs,i}} - E_{p_{est,i}})^2}{\sum_{i=1}^{n}(E_{p_{est,i}} - E_{p_{est,m}})^2}, \tag{8}$$

$$SI = \sqrt{\frac{\sum_{i=1}^{N}[(E_{p_{est,i}} - E_{p_{est,m}}) - (E_{p_{obs,i}} - E_{p_{obs,m}})]}{\sum_{i=1}^{N}E_{p_{obs,i}}^2}}, \tag{9}$$

where $E_{p\ obs,i}$, $E_{p\ est,i}$ observed and estimated pan evaporation values on the $i^{th}$ day, $E_{p\ obs,m}$ and $E_{p\ est,m}$ is the mean value of $E_p$ obs,i and $E_{p\ est,I}$, respectively. The total number of testing patterns is denoted by n and i represent the number of particular instances of the testing pattern.

## 3 Results

### 3.1 Meteorological variables and pan evaporation

The long-term (1971–2000) growing season's average $T_a$ at Keszthely is 18.8 °C, the hottest month is July with a mean monthly $T_a$ of 20.5 °C, while the coolest month is September (15.7 °C). In the study period, the seasonal mean $T_a$ were 5.5-15.7% higher than the 30-year average. Out of six seasons studied, three warm (2015, 2017, 2019) and three close to normal (2016, 2018, 2020) ones could be distinguished (Fig 4). The seasonal mean $T_a$ in warm seasons were 11.5%-15.7% higher than that of the climate norms.

The climate of Keszthely is characterized by highly variable and irregular P with a long-term seasonal total of 274.3 mm from June to September. Monthly seasonal mean precipitation sums varied from 78.5 mm (June) to 57.1 mm (September). Warm seasons (2015, 2017, 2019) were characteristically arid with 4.9-21.6% less seasonal total P, respectively, compared to the 30-year average. In the other study seasons, there were 23.9-40.4% more P (data not shown) than that of the climate norm.

Figure 4 displays the meteorological variables and observed daily $E_p$ in different pan treatments determined in a box-and-whisker plot between growing season 2015-2020, indicating minimum, first quartile, median, third quartile, and maximum values. An increasing trend was observed in the $T_{min}$ with an increment of 9.6% while the $T_{max}$ exhibited an unchanged trend over the studied growing seasons. In the study location, there were hardly any differences in seasonal mean RH values (0.6-

9.2%) and daily $R_s$ sums (21.3-24.3 W m$^{-2}$) between 2015 and 2020. The highest (1.6 m s$^{-1}$) and lowest (0.9 m s$^{-1}$) seasonal mean wind speeds were measured in 2016 and 2018, respectively.

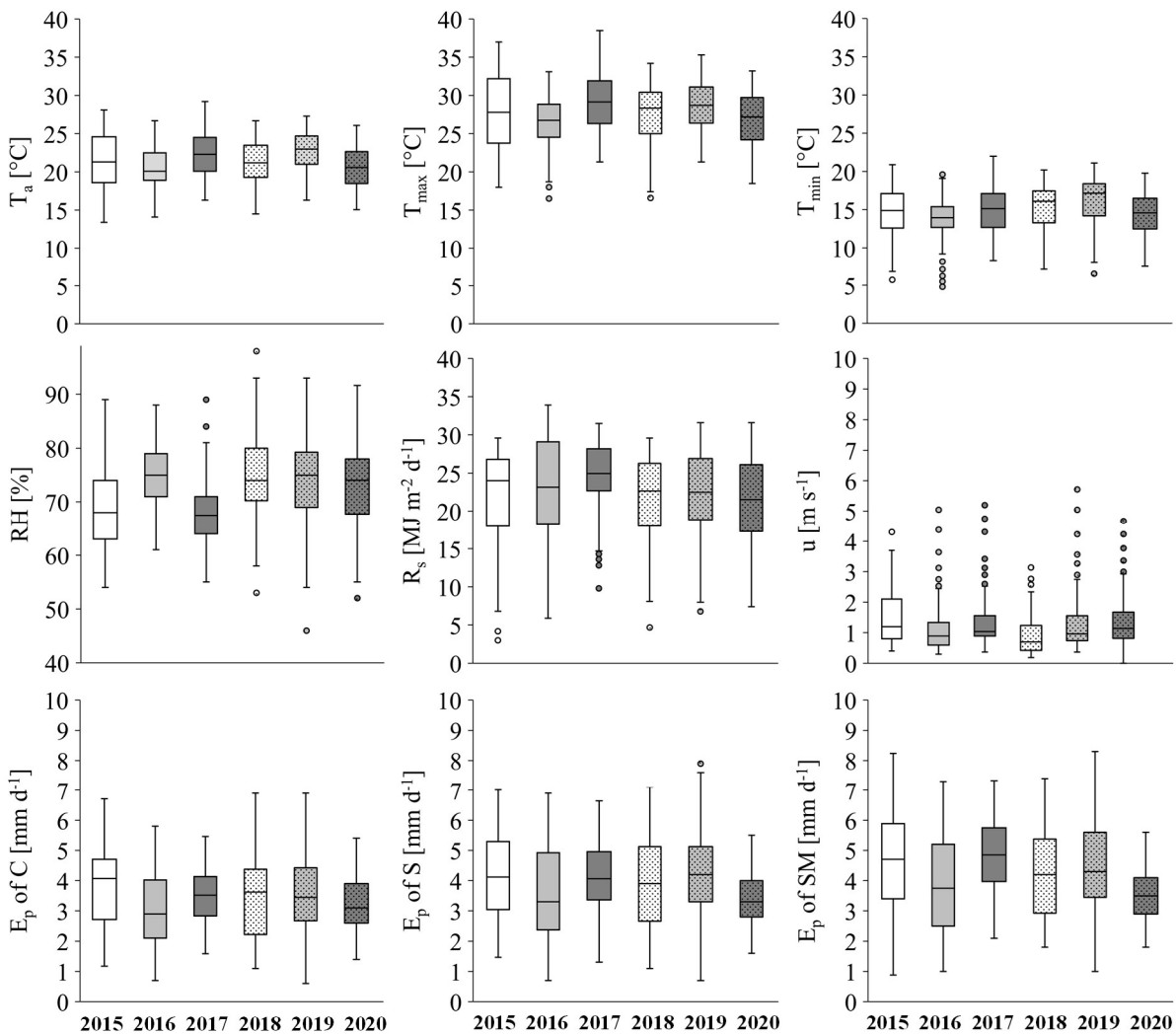

**Figure 4: Box plot of meteorological parameters (T$_a$ – daily mean temperature [°C], T$_{max}$ – daily maximum temperature [°C], T$_{min}$ – daily minimum temperature [°C], RH – relative humidity [%], R$_s$ – global radiation [MJ m$^{-2}$ d$^{-1}$], u – wind speed [m s$^{-1}$]) and daily evaporation of different pan treatments [mm d$^{-1}$] (C – control, S – Class A pan with sediment cover bottom, SM – Class A pan with submerged macrophyte) in 2015-2020 growing seasons (June-September). The lower and upper ends of the box indicate the 25$^{th}$ and 75$^{th}$ percentiles of the variances, respectively, while the horizontal bar within the box indicates the median. The two horizontal bars**
**indicate the range that covers 90% of the variances. Outliers are indicated with circles.**

Daily $E_p$ rates were related to seasonal $T_a$ variations and not to rainfall patterns. Higher daily mean water losses were registered during the warm-dry seasons (C: 3.5-3.8 mm day$^{-1}$, S: 4.2-4.3 mm day$^{-1}$, SM: 4.5-4.9 mm day$^{-1}$), while somewhat lower average $E_p$ rates were measured in the three normal seasons (C: 3.0-3.5 mm day$^{-1}$, S: 3.4-4.0 mm day$^{-1}$, SM: 3.6-4.2 mm day$^{-1}$). As a result of pan seeding, differences in daily mean $E_p$ rates were more pronounced in warm summers. In warm seasons, significant

deviations of daily mean $E_p$ between C and S (p<0.001) as well as S and SM (p<0.001) were observed. At the same time, significant differences in daily mean $E_p$ between C and S (p<0.001) and C and SM (p<0.001) were registered in normal seasons. No significant impact of pan seeding in all the remaining treatments was detected (p=0.0693-0.0896) (Figure 4). A two-way ANOVA was conducted to explore the impact of the studied seasons and the treatment on $E_p$ rates. There were significant main effects caused by the growing season ($F_{(5, 211)}$ = 24.241, p = 0.001) and the pan treatment ($F_{(2, 236)}$ = 67.855, p = 0.001) in full dataset. The interaction between seasons and treatments was not significant ($F_{(10, 29)}$ = 0.085, p = 0.503). Tukey HSD post-hoc tests revealed significant differences among the three pan treatments (p < 0.001 for all pairwise comparisons) for the training, testing phase and full dataset (Table 1).

Table 1. The impact of sediment (S) and submerged aquatic macrophytes (SM) on evaporation rates ($E_p$) of Class A pan (C) in the full data set (2015-2020), training (2015-2017) and testing (2018-2020) phase with 95% confidence intervals

| Multiple Comparisons | | | | | | |
|---|---|---|---|---|---|---|
| | | | | | 95% Confidence Interval | |
| | | Mean difference | | | | |
| (I) treatment | (J) treatment | (I-J) | Std. Error | Sig. | Lower Bound | Upper Bound |
| Full dataset (2015-2020) | | | | | | |
| C | S | -0.490* | 0.0733 | **0.000** | -0.662 | -0.318 |
| | SM | -0.845* | 0.0735 | **0.000** | -1.017 | -0.672 |
| S | C | 0.490* | 0.0733 | **0.000** | 0.318 | 0.662 |
| | SM | -0.355* | 0.0733 | **0.000** | -0.526 | -0.183 |
| SM | C | 0.845* | 0.0735 | **0.000** | 0.672 | 1.017 |
| | S | 0.355* | 0.0733 | **0.000** | 0.183 | 0.526 |
| | | Based on observed means. The error term is Mean Square (Error) = 1.741. | | | | |
| Training data set (2015-2017) | | | | | | |
| C | S | -0.712* | 0.1066 | **0.000** | -0.962 | -0.462 |
| | SM | -0.731* | 0.1072 | **0.000** | -0.982 | -0.479 |
| S | C | 0.712* | 0.1066 | **0.000** | 0.462 | 0.962 |
| | SM | -0.019* | 0.1124 | **0.019** | -0.283 | 0.245 |
| SM | C | 0.731* | 0.1072 | **0.000** | 0.479 | 0.982 |
| | S | 0.019* | 0.1124 | **0.019** | -0.245 | 0.283 |
| | | Based on observed means. The error term is Mean Square (Error) = 1.840. | | | | |
| Testing data set (2018-2020) | | | | | | |
| C | S | -0.505* | 0.0993 | **0.000** | -0.738 | -0.272 |
| | SM | -0.716* | 0.1001 | **0.000** | -0.951 | -0.481 |
| S | C | 0.505* | 0.0993 | **0.000** | 0.272 | 0.738 |
| | SM | -0.211* | 0.0990 | **0.045** | -0.443 | 0.022 |
| SM | C | 0.716* | 0.1001 | **0.000** | 0.481 | 0.951 |
| | S | 0.211* | 0.0990 | **0.045** | -0.022 | 0.443 |

*The mean difference is significant at the 0.05 level.

The correlation of evaporation of different pan treatments with other meteorological variables is also given in Table 2. There was a statistically significant difference in evaporation rates of full datasets as well as in the case of training and testing datasets between the seeded and classic Class A pan. The $T_a$, $T_{max}$ and $R_s$ positively impacted the $E_p$, while RH had a negative correlation with $E_p$. In this study, u hardly affected the $E_p$ rates irrespective to treatment. The descriptive statistics of both training and testing datasets showed that most of the meteorological variables and $E_p$ were similar to the full data set.

Table 2. Statistics of meteorological variables ($T_a$ - mean air temperature, $T_{max}$ - maximum air temperature, $T_{min}$ - minimum air temperature, RH - relative humidity, $R_s$ - solar radiation, u - wind speed) and their correlation with evaporation ($E_p$) of C, S and SM in the full time series (2015-2020), training (2015-2017) and testing phases (2018-2020). C, S and SM are control class A pan, A pan with sediment cover-bottom and A pan with planted freshwater submerged macrophyte, respectively.

| Data set | Statistics | $T_a$ [°C] | $T_{max}$ [°C] | $T_{min}$ [°C] | RH [%] | u [m s⁻¹] | $R_s$ [MJ m⁻² day⁻¹] | $E_p$ of C [mm d⁻¹] | $E_p$ of S [mm d⁻¹] | $E_p$ of SM [mm d⁻¹] |
|---|---|---|---|---|---|---|---|---|---|---|
| Full (2015-2020) | Average±SD | 21.1±3.2 | 27.5±4.0 | 14.8±3.2 | 72.7±8.0 | 1.3±0.9 | 22.4±6.0 | 3.4±1.2 | 3.9±1.4 | 4.3±1.5 |
| | Correlation with $E_p$ of C | 0.59** | 0.53** | 0.42** | -0.43** | 0.01 | 0.50** | 1.00 | - | - |
| | Correlation with $E_p$ of S | 0.57** | 0.51** | 0.40** | -0.42** | 0.03 | 0.53** | 0.92** | 1.00 | - |
| | Correlation with $E_p$ of SM | 0.56** | 0.50** | 0.37** | -0.44** | 0.01 | 0.52** | 0.90** | 0.93** | 1.00 |
| Training (2015-2017) | Average±SD | 20.9±3.4 | 27.5±4.4 | 14.4±3.3 | 71.0±7.5 | 1.4±0.9 | 23.1±6.1 | 3.4±1.2 | 4.0±1.4 | 4.4±1.6 |
| | Correlation with $E_p$ of C | 0.65** | 0.59** | 0.49** | -0.48** | 0.05 | 0.51** | 1.00 | - | - |
| | Correlation with $E_p$ of S | 0.63** | 0.58** | 0.45** | -0.47** | 0.00 | 0.56** | 0.91** | 1.00 | - |
| | Correlation with $E_p$ of SM | 0.63** | 0.57** | 0.44** | -0.50** | 0.04 | 0.54** | 0.89** | 0.93** | 1.00 |
| Testing (2018-2020) | Average±SD | 21.2±2.9 | 27.4±3.5 | 15.3±3.0 | 74.2±8.2 | 1.2±0.9 | 21.8±5.7 | 3.4±1.2 | 3.9±1.4 | 4.1±1.4 |
| | Correlation with $E_p$ of C | 0.53** | 0.46** | 0.35** | -0.41** | 0.06 | 0.51** | 1.00 | - | - |
| | Correlation with $E_p$ of S | 0.51** | 0.44** | 0.35** | -0.39** | 0.06 | 0.50** | 0.92** | 1.00 | - |
| | Correlation with $E_p$ of SM | 0.49** | 0.41** | 0.33** | -0.38** | 0.05 | 0.49** | 0.92** | 0.95** | 1.00 |

On the basis of the daily variation of $T_w$ in different depths, two time-periods were distinguished (Figure 5); daytime (7:00 – 18:00 h, LMT) and nighttime cooling (19:00 – 6:00 h, LMT). With clear sky conditions, the surface Tw peaked at 14:00 h, irrespective to treatment. The magnitudes of surface Tw in daytime (between 07:00 and 14:00 hours) increased from 21.6 to 37.5ºC in C, from 23.0 to 37.4ºC in S, and from 19.8 to 38.0ºC in SM. Then, with declining solar radiation, the Tw slightly decreased during the nighttime cooling to 21.2, 21.8 and 18.7ºC in C, S and SM, respectively, until sunrise. In deeper water

depth, a similar pattern of Tw with slightly smaller magnitudes was measured with a time lag of 1-to-2-hours from the surface Tw. In the classic A pan, the Tw in deeper depth from the surface did not reduce as rapidly as Tw in seeded pans. On cloudy days, insignificant Tw differences less than 1ºC (p=0.059 - 0.969) between the neighbouring layers were observed in every treatment.

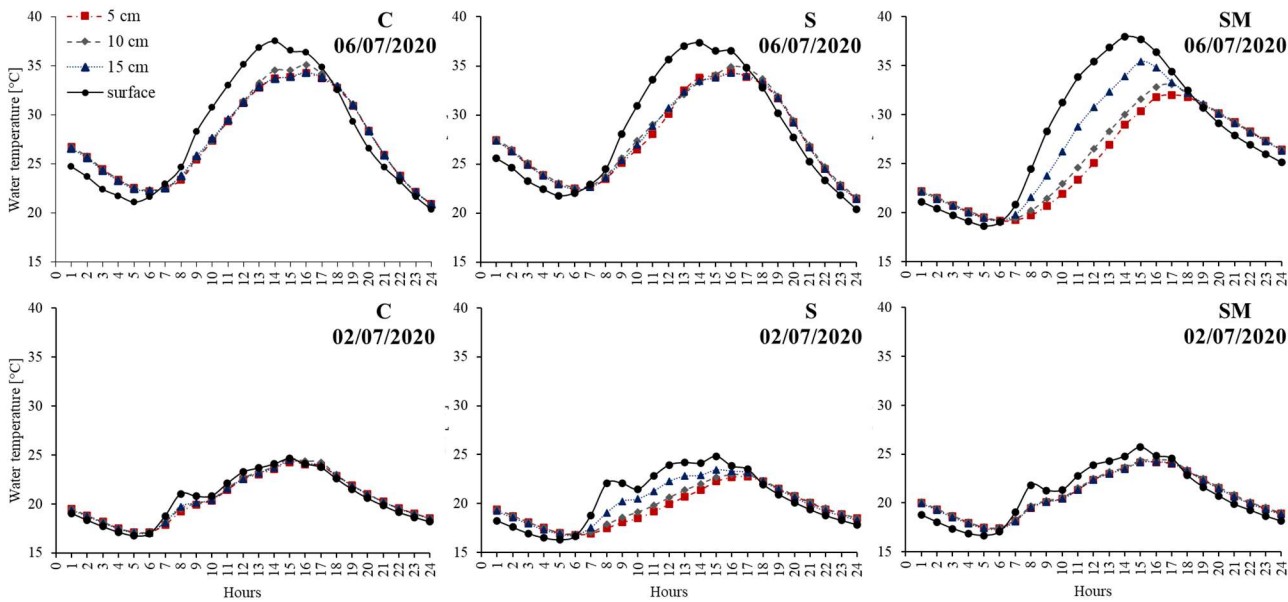

**Figure 5: Water temperature of different pan treatments (C – Class A pan/control; S – Class A pan with sediment covered bottom; SM – Class A pan with submerged macrophyte) in clearsky and cloudy sample days. The layers represent the distance from the pan bottom. The lowest sensor' height was 5 cm.**

### 3.2 K-SOM features

Table 3 shows the usual parameter table for K-SOM. The following steps were required to present Fig. 5: inputs were normalized, the code book was generated, the map size complied with the dimensions of the component planes. The neighbouring function of the pixels was Gaussian, the shapes of component planes were sheets, and the planes shapes were

hexagonal. Two indicators are most often used to qualitatively evaluate the two main goals of the K-SOM algorithm: quantization error (QE) and topographic error (TE) (Table 3). The QE shows how closely the map vectors match the data vectors, thereby quantifying map resolution (Kohonen, 1995). The TE, in turn, determines the extent to which the topology of the input data structure is preserved on the output map (Kiviluoto, 1996). QE and TE does not have a default value, but the smaller the QE and TE (if the values tend to be zero) the better the model is. In this study, the values of QE and TE were equal to 0.016 and 0.820, respectively, indicating that the K-SOM was appropriately trained in topology.

Table 3 Characteristics of trained Kohonen Self-Organizing Map (K-SOM) model

| Characteristics | Values |
| --- | --- |
| Normalization method | variance: $x' = (x - \bar{x})/\sigma_x$ |
| Codebook | 312 x 3 |
| Map Size | 24 x 13 |
| Neighbourhood function | Gaussian |
| Shape | Sheet |
| Lattice | Hexagonal |
| Final Topographic error (TE) | 0.820 |
| Final Quantization error (QE) | 0.016 |

K-SOM can be interpreted using the output map and the individual component planes, so the relationships between each variable can be explored. The component planes help to visually illustrate areas in which the intensity of the relationship of the variables is high, low, or average and thus helps to better understand the relationship between the $E_p$ and meteorological variables. The component planes for each variable of the K-SOM model are shown in Figure 6. Superimposed on K-SOM patterns of input meteorological variables, radiation, air temperatures including minimum and maximum, relative humidity, wind speed could be captured revealing their co-variability with the pan evaporation.

In the map, the similar weight vectors shave similar colours, based on the U-matrix according to a naïve contraction model proposed by Himberg (2000) and Peeters et al. (2007). Among NN features, as the clustering, classification, prediction, and data mining in large datasets (Kohonen and Somervuo, 2002; Kalteh et al., 2008) only the prediction and data mining were applied in the study. As there was no group distinction (classification), the U-matrix has not been presented here.

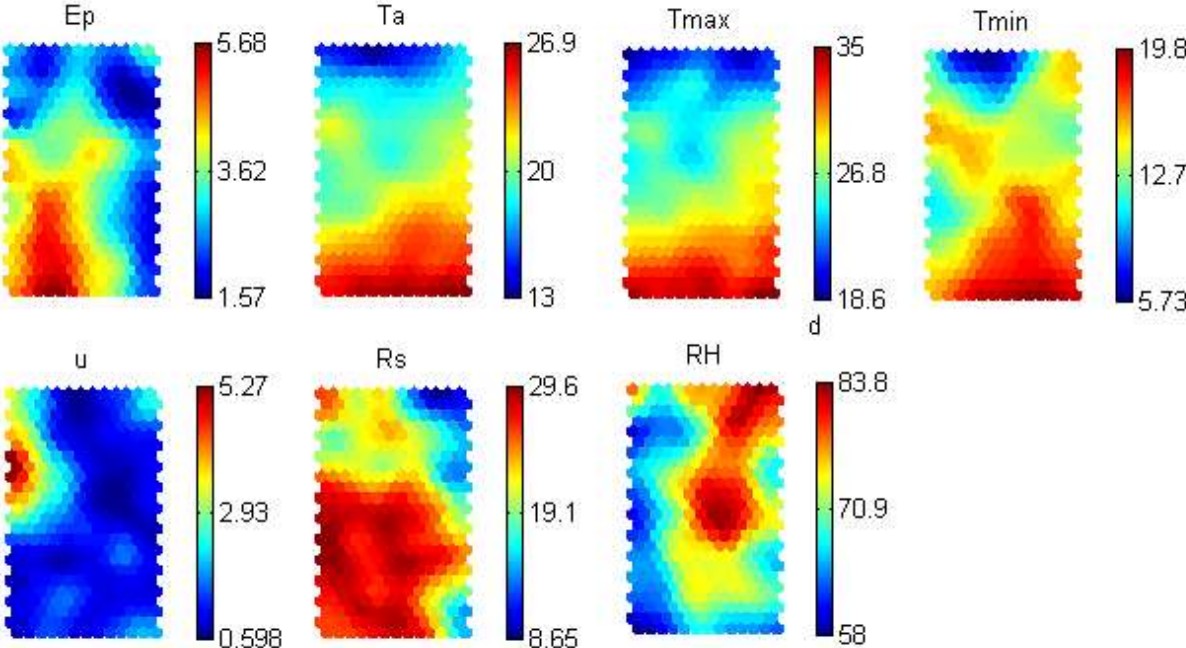

**Figure 6: Kohonen Self-Organizing Map (K-SOM) visualization of pan evaporation and meteorological variables assessment ($T_a$ – daily mean temperature [°C], $T_{max}$ – daily maximum temperature [°C], $T_{min}$ – daily minimum temperature [°C], RH – relative humidity [%], $R_s$ – global radiation [MJ m$^{-2}$ d$^{-1}$], u – wind speed [m s$^{-1}$] and $E_p$ - daily evaporation [mm d$^{-1}$]). The bars indicate the intensity of the variables: the red colour is high importance, and the blue colour is low importance.**

A colour was assigned to a node in accordance with the relative value of the respective component in that node (Li et al., 2018). On the maps, the warm colours (red, orange) show positive correlation between the study variables, and the darker the colours (blue), the lower the relative value of the component of the corresponding variable. When one variable is red, while the other one is blue on the same place of the heat map, the correlation between them will be negative. Thus, the correlation between the K-SOM modelled values of $E_p$, $T_a$, $T_{min}$, $T_{max}$, $R_s$, RH, and u becomes clearly visible. The colour gradient of $E_p$ was similar to those for variables related to available energy ($T_a$, $T_{min}$, $T_{max}$ and $R_s$), indicating that these contribute most to the increase of $E_p$. The component planes also visually confirm the negative correlation between RH and $E_p$, with high values of the RH resulting in low values of the $E_p$.

**3.3 FAO56-PM, MLR and K-SOM models**

Figure 6. depicts the time variation and X-Y scatter plots of the observed and estimated daily $E_p$ values obtained by C, S and SM during the testing period (2018-2020).

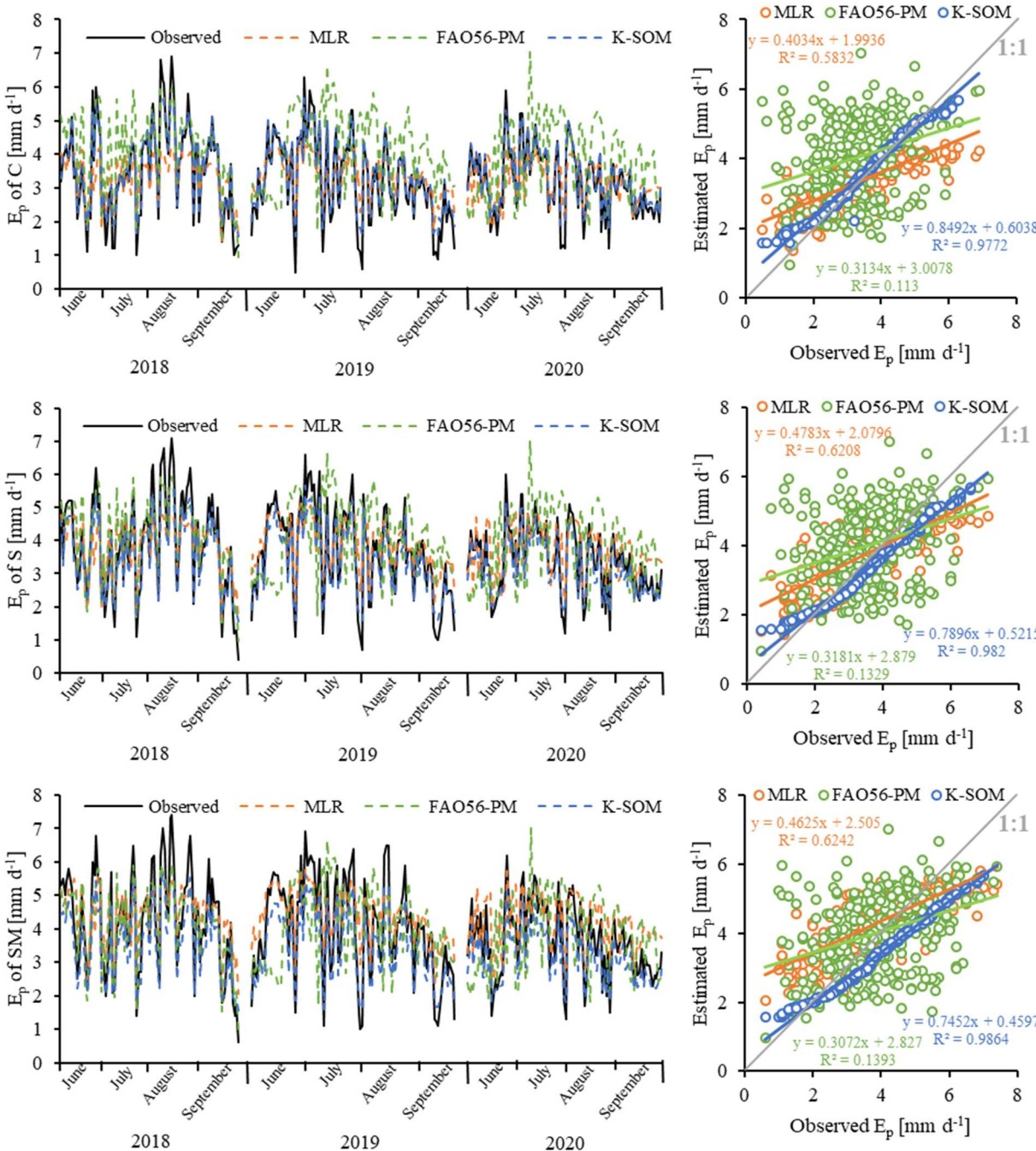

**Figure 7: Time series and X-Y scatter plot of observed and predicted daily pan evaporation (E$_p$) in different pan treatment (C – control, S – pan with sediment cover bottom, SM – pan with submerged macrophytes) by daily multiple stepwise regression (MLR), FAO-56 Penman-Monteith (FAO56-PM) and Kohonen self-organization map (K-SOM) models during testing period (2018-2020) growing seasons). All probability levels were equal to p<0.001.**

From the Figure 7., it can be observed that most of the estimated daily $E_p$ values (for MLR and K-SOM) are close to the observed daily $E_p$ values for all three pan treatments. The possible reason of low $R^2$ values in FAO56-PM might be the role of the variable that is the estimate in crop potential evapotranspiration and not evaporation in water bodies. The regression line is above the 1:1 line up to 4 mm, which means that the FAO56-PM and MLR models slightly overestimated the magnitude of the daily $E_p$ values in different pan treatments. However, above 4 mm daily $E_p$, the FAO56-PM and MLR models already underestimated the observed $E_p$ values. The daily $E_p$ values of C, S, and SM of the K-SOM model follow the 1:1 line most accurately. For all three models, $R^2$ values were highest for SM treatment (FAO56-PM: 0.1393, MLR: 0.6242, K-SOM: 9864). In the case of K-SOM, it can also be observed that low $E_p$ values are overestimated, while higher $E_p$ values are underestimated, although the estimated "middle" $E_p$ values (which occur most frequently in a growing season) were close to the observed $E_p$ values regardless of pan treatment. A greater degree of underestimation is observed for SM treatment for K-SOM.

In this study, we developed $E_p$ models based on three different approaches (FAO56-PM, MLR and K-SOM) with daily meteorological variables, and tested the performance of the models by four commonly used statistical indicators (MAE [Ideal = 0, (0,+∞)], RMSE [Ideal = 0, (0,+∞) ], NSE [Ideal = 1, (−∞,1)], SI [Ideal = 0, (0,+∞)]). Figure 8 shows the overall performance of the three predicted methods at the three pan treatments.

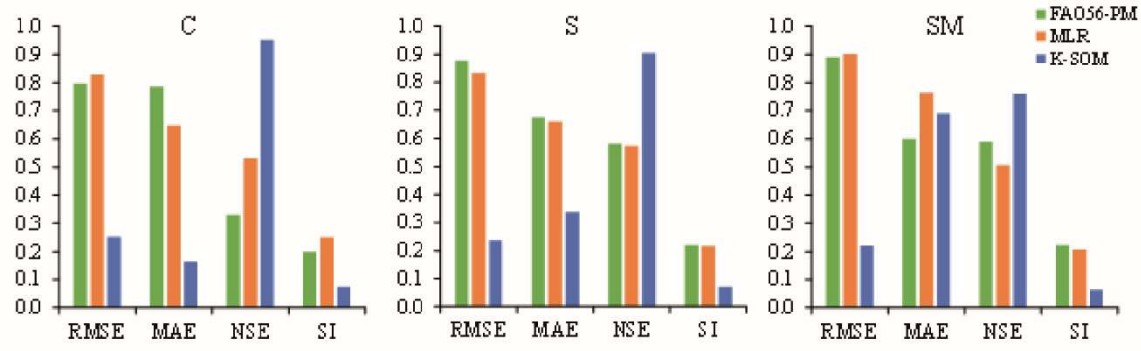

**Figure 8: Error statistics (root mean square error - RMSE, mean absolute error - MAE, scatter index - SI and Nash-Sutcliffe efficiency - NSE) for the multiple stepwise regression (MLR), FAO-56 Penman-Monteith reference crop evapotranspiration (FAO56-PM) and Kohonen self-organization map (K-SOM) models during the testing period for different pan treatments (C is standard class A pan with clean water, S is class A pan with sediment cover bottom, and SM is class A pan containing submerges macrophyte).**

The K-SOM models (RMSE = 0.222–0.253; NSE = 0.761–0.951; SI = 0.065–0.074) performed the best in the testing period, and their RMSE and MAPE were lower, and their NSE were higher than those of FAO56-PM and MLR models regardless of pan treatment (C: 0.951; S: 0.906; SM: 0.761). Additionally, the MAE value for treatments C and S was the lowest in the K-SOM models (MAE = 0.164 and MAE = 0.338, respectively), in contrast, the FAO56-PM had the best MAE value for SM treatment (MAE=0.601).

Overall, the MLR (RMSE = 0.834; MAE = 0.660; S = 0.217) was slightly superior to FAO56-PM (RMSE = 0.877; MAE = 0.675; SI = 0.220) in the S, and there was only a small difference in the value of NSE between the two models (MLR: 0.572; FAO56-PM: 0.580). In the C treatment, RMSE (0.796) and SI (0.200) were lower for FAO56-PM, while MAE (0.648) and

NSE (0.531) values were more favourable for the MLR model. Nevertheless, both the K-SOM model and MLR model were better than the FAO56-PM model during the testing period for "non-empty" treatments (S and SM).

## 4 Discussion

To date, there is little information about the impact of submerged aquatic macrophytes on $E_p$ rate. According to a previous
study in India (Kota, Rajasthan), water hyacinth evapotranspirated 26% more water than free water surface in a 9-month experiment (Brezny et al., 1973). In the same place as this study, Anda et al. (2016; 2018) have shown that the presence of sediment increases the evaporation of the Class A pans by an average of 12.7% and the submerged aquatic macrophytes by an average of 21.3%, between 2014 and 2016,. Jiménez-Rodríguez et al., (2019) reported that the observed $E_p$ were higher for aquatic plants than the open water cover in Palo Verde National Park, Costa Rica, between December 2012 and January 2013
(45 days). Concerning the relationship between pan treatments and meteorological variables, it can be concluded that positive correlation was observed with most meteorological variables, while a negative correlation was observed with RH. This result was supported by other studies in the literature (Sheffield et al., 2017). In this study, u hardly affected the $E_p$ rates of each treatment. This does not confirm the conclusions made by earlier studies (McVicar et al., 2012). This may be due to the fact that Keszthely is sheltered by surrounding mountains causing lower wind speeds (Anda et al., 2016).
Daily mean $T_w$ increases were 5.4 and 4.5ºC in S and SM, respectively, compared to C during clear-sky conditions. Despite the less intense stratification on overcast days, $T_w$ of seeded pans was 5.4ºC higher than that of daily mean $T_w$ of C.

In accordance with shallow lake stratification results of Jacobs et al. (1998), increased stratification was evident in daytime, but the number of layers strongly depended on macrophyte presence. More moderate $T_w$ layer differences were also present at night. The stratification was the most intense with 3 significantly different layers (p<0.001) in seeded pans, during clear-sky
daytime. At the same time, the number of layers with varied $T_w$ was only 2 (p<0.001 – p=0.012) in classic A and sediment covered pans. Results in the study were confirmed by Andersen et al. (2017) concluding that shallow lakes colonized by submerged macrophytes strongly stratify the water body, mainly during the daytime. The reason of this stratification is the dissipating turbulent kinetic energy and absorbing heat (Vilas et al., 2018). The plants may act as a barrier to seeded pans water mixing, attenuating underwater light, thereby enhancing the thermal stratification inside the pan's water column.
The strength of stratification, the daily mean $T_w$ differences between the surface and bottom water were 2.5 (p=0.005), 3.0 (p<0.001) and 6.5ºC (p<0.001) in C, S and SM, respectively, on cloudless days. At night-time cooling, variation in $T_w$ between different layers was less pronounced, remaining below 1ºC (p<0.001 – p= 0.005).

In addition to stratification, the macrophytes have strengthened the daily variation of $T_w$ in different depth. A 0.3ºC increase in daily mean surface $T_w$ of seeded pans related to C was obtained during daytime, with a variation ($T_{max} - T_{min}$) of 18.4 and
375 19.3ºC in C and SM, respectively. On the bottom, an opposite trend in daytime mean $T_w$ was detected; the seeded pans $T_w$ in 0.05 m depth was 3.1ºC (p=0.040) cooler than that of the $T_w$ of C. Probably the macrophyte presence resulted in insufficient downward heat transport, maintaining the more stratified water body of seeded pans. Herb and Stefan (2004) also found

reduced turbulent mixing in shallow Otter Lake, Minnesota, with rooted macrophytes. The authors observed that $T_w$ fluctuations at 20 cm depth were 3ºC in open water and 4.5ºC in lake water with macrophyte cover. Evapotranspiration functions of SM fitted to surface $T_w$ evolution; the higher the surface $T_w$, the more intense the $E_p$ rate was measured in SM related to $E_p$ of classic A pan.

Many researchers have conducted research with neural networks aimed at the estimation of $E_p$ as a function of meteorological variables (Keskin and Terzi, 2006). Several of these researchers found better results in $E_p$ estimation with neural network approach than those obtained from the Priestley-Taylor and the Penman methods (Rahimikhoob, 2009; Malik et al., 2020b). Consistent with other studies, this study demonstrated that modelling of $E_p$ is possible through the use of K-SOM technique in addition to the FAO56-PM and MLR methods. The comparison results indicated that, in general, the K-SOM model was superior to the FAO56-PM and MLR methods. Chang et al. (2010) used different methods to estimate pan evaporation, including also the K-SOM and the FAO56-PM. According to the results of Chang et al. (2010), K-SOM was the best of the studied methods, and it was found that the Penman-Monteith method is also likely to underestimate evaporation. Malik et al. (2017) used four heuristic approaches and two climate-based models to approximate monthly pan evaporation, where the K-SOM model performed better than the climate-based models. The regression line in scatter plots has $R^2$ as 0.937 for K-SOM model at Pantnagar and Ranichauri (India). In the study of Malik et al. (2017), RMSE values were 0.685 and 1.126 for K-SOM, when 50% of the total available data was used in the testing of models in two stations.

**4 Conclusions**

The $E_p$ of a class A pan with submerged aquatic macrophytes and with a sediment-covered bottom was observed at Keszthely, over six consecutive (2015-2020) growing seasons. In this study, it was attempted to model $E_p$ by employing models consisting of FAO56-PM, MLR and K-SOM, using daily pan evaporation values in different Class A pan treatments (C, S, SM). The $E_p$ rate of SM and S was always significantly higher than that of the "empty" class A pan each growing season. The presence of submerged macrophyte resulted in a higher $E_p$ than in the sediment-covered class A pan.

Macrophyte induced thermal stratification in water bodies (lakes/evaporation pans) emerge only in the vegetation period, during macrophyte development. One less layer in Classic A pan compared to macrophyte seeded pans was probably due to modified $T_w$ stratification causing changed water column stability. Wider $T_w$ values induced dynamics presented in the macrophyte seeded pans demonstrated the possibility of developing a more heterogenous environment for aquatic ecosystems. Macrophyte induced modified thermal stratification with higher surface $T_w$ could explain the increased $E_p$ in seeded pans.

Modified $E_p$ of seeded pans made those values closer to the $E_p$ of natural lakes with submerged macrophytes. While the $T_w$ stratification trend in SM was similar to that of natural shallow lakes, it may also provide a new consideration for routine hydrometeorological management. $T_w$ distribution in macrophyte covered lakes impacts other physical properties such as nutrient cycling, dissolved oxygen, etc. When treating $E_p$ from a pan to that from a vegetated surface including lakes or other

aquatic habitats, to improve evaporation estimation, multidimensional approximation is necessary, offering simple methods

for end-users including hydrologists, meteorologists, or any other specialists.

Daily $E_p$ rates for all pan treatments were related to seasonal $T_a$ variations. Correlation analysis revealed that $T_a$, $T_{max}$, $T_{min}$ and $R_s$ had a positive correlation with pan evaporation, whereas RH had a negative correlation (-0.42 to -0.44) with $E_p$ of C, S and SM in full dataset. Among all, the R (correlation coefficient) of $T_a$ (ranged from 0.56-0.59) had a stronger positive correlation followed by R of $T_{max}$ (ranged from 0.50-0.53) and R of $R_s$ (ranged from 0.50-0.53). The relationship with u was low for the

$E_p$ of the three treatments, which can be explained by the low u of Keszthely in the growing seasons. Using the visualization capability of the K-SOM, it was clearly confirmed that the $E_p$ was more closely correlated with the variables related to available energy, than the RH.

The performance accuracy of the different applied models was evaluated with RMSE, MAE, NSE and SI statistics. Results showed that the K-SOM model has accuracy in prediction precision over the FAO56-PM and MLR models. Comparing the

FAO56-PM and MLR models, MLR performed better in this study in S and SM treatments.

Since the $E_p$ of one sample place was included in the study, the 'generic' impact of submerged macrophytes on $E_p$ was not fully discussed; maybe for different reasons, our results in other sites became variable. More surveys are needed to reveal the applicability of planted standard A pan $E_p$ for different geographical and climatic conditions.

A possible application value of the study is in validating the presence of littoral sediments and macrophytes in evaporation

estimation, the amount of lost water by wetlands that can easily be accounted in the prediction of their performance. Results from the study may also contribute to the protection of aquatic plants and to environmental management of wetlands also in other regions of the world. Management strategies aiming to estimate accurate water budget terms including evaporation can be a realistic aim for preventing further inaccurate water loss projections.

**Acknowledgements**

Project PD_21 was implemented with the support of the Ministry for Innovation and Technology from the Source of the National Research, Development and Innovation Fund, in the financing of the PD 138660 application program.

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
