# Peer review of "Pan evaporation is increased by submerged macrophytes"

_Hydrology and Earth System Sciences, 2021_

## Referee Comment (RC2)

"Estimation Standard And Seeded Pan Evaporation Using Modelling Approach" by Brigitta Simon-Gáspár et al.

General comments

The paper concerns a topic consistent with the research domain of the Special issue: Experiments in Hydrology and Hydraulics in the HESS journal.

The authors used FAO-56 Penman-Monteith equation (FAO56-PM), multiple stepwise regression (MLR) and Kohonen self-organizing map (K-SOM) techniques to estimate daily pan evaporation (Ep) in three treatments. And in an six-season experiment. The 10 modelling approach included six measured meteorological variables were compared and evaluated. The results showed that the MLR method provided close compliance with the observed pan evaporation values, but the K-SOM method gave better estimates than the other methods.

I really appreciate the huge work made by the authors. However, I have some questions about the innovation of the research method and the purpose of this study. First, the ,methods of FAO-56 Penman-Monteith equation (FAO56-PM) (Allen et al., 1998), Kohonen self-organizing map (K-SOM) techniques (Kohonen, 1982) and multiple stepwise regression (MLR) are classic, but also relatively old methods. Predecessors have done a lot of research and published a lot of relevant papers on the comparison of these methods. If only a few traditional, classical methods are compared and discussed, I think that there is not enough innovation in terms of methodology to be published in HESS, and there are already many ready-made papers on comparative studies of evaporation calculation methods. In addition, the authors said that "there is little information in the literature on how submersed macrophytes affect the evaporation of a lake" in Introduction (Lines 64). Is this statement supported by the literature? (Wang, J.H. 1994. Effects of aquatic plants on water surface temperature and evaporation. Arid

Land Geography, 17(2), 3. doi: CNKI:SUN:GHDL.0.1994-02-009).

The second main aim of this paper was to estimate daily Ep using FAO-56 Penman-Monteith (FAO56-PM), Kohonen self-70 organization map (K-SOM) and multiple stepwise linear regression (MLR) methods. Since this purpose is only the comparison and evaluation of several traditional methods, and I personally still feel that the innovation is not sufficient for HESS.

Specific Comments

1. English language needs to be modified. I found several unclear sentences that make it difficult to understand the analysis and results carried out.

2. The description at the beginning of the Abstract is too simple and empty, two to three sentences should be used to focus on the shortcomings of the current study and the innovation of this study.

3. Some of the references in the Introduction are too old. It is suggested that the author update some relevant studies recently published.

4. The font resolution in Figure 1 is too low to see the relevant text clearly. I suggest the author to redraw it.

5. The numeric font in equation 2 is suggested to be Times New Roman, and the rest of the formula is the same.

6. In Materials and methods, it would be better to give specific steps about the experimental design of this study, the current presentation is relatively sketchy.

7. The Results and Discussion session. I found some good results from this study, but unfortunately the authors' description of these results is too brief (Both in Figures and Tables)and suggest a more specific analysis and evaluation of the results. And the discussion was not in-depth enough and was only a brief description of the Results. It is recommended to fully evaluate and discuss the results obtained by several different methods used in this study in terms of the mechanism of influence.

8. The content of the conclusion should not be a simple retelling of the results and discussion, but also a more in-depth explanation of the scientific significance and potential application value of the study, rather than the kind of formulaic statement in the last paragraph of the conclusion.

9. I am not quite sure if the current hess format requires line numbers to be marked every five lines, which causes some reading difficulties, if not required by the journal format, it is recommended that authors mark all line numbers.

---

## Referee Comment (RC4)

**TITLE:** Estimation Standard And Seeded Pan Evaporation Using Modelling Approach

**Reviewers' comments:**

In this paper, K-SOM and other methods are used to improve the estimation of lake evaporation, which is conducive to accurately estimate the total lake evaporation and improve the climate effect of the lake under the background of climate change. I recommend publication of the paper in HESS after revision.

**Major comments**

Has this article been studied by simulation at Keszthely, Hungary. However, how do you consider the effect of the evaporation of non-uniform underlying surfaces, such as mountains, and grass?

This paper improves the calculation method of lake evaporation, and further analysis of lake evaporation and its climate effects are needed on the Lake Balaton in the future.

**Minor comments**

Figure 1 should be topography.

L.47 The unsupervised NNs, including Kohonen Self Organizing Maps (K-SOM), has several advantages (Kohonen, 1982). Full name should be given for the first occurrence 'NN'.

L.16-20 Performances of the different models were compared using statistical indices, which included the root mean square error (RMSE), mean absolute error (MAE), scatter index (SI) and Nash-Sutcliffe efficiency (NSE). The results showed that the MLR method provided close compliance with the observed pan evaporation values, but the K-SOM method gave better estimates than the other methods. Overall, K-SOM has high accuracy and huge potential for Ep estimation for water bodies 20 where freshwater submerged macrophytes are present. This section need to be rewrite.

L.84 (latitude: 46°44′N, longitude: 17°14′E, elevation: 124 m above sea level) 'above sea level' Can be abbreviated as a.s.l.

L.231 From the figure, it can be observed that most of the estimated daily Ep values are close to the observed daily Ep values for all three pan treatments.
Which figure?

Many researchers have conducted research with neural networks aimed at the estimation of Ep as a function of meteorological variables (Keskin and Terzi, 2006). Several of these researchers found better results in Ep estimation with neural network than those obtained from the Priestley-Taylor and the Penman methods (Rahimikhoob, 2009; Malik et al., 2020). Consistent with other studies, this study demonstrated that modelling of Ep is possible through the use of K-SOM technique in addition to the 275 FAO56-PM and MLR methods. The comparison results indicated that, in general, the K-SOM model was superior to the FAO56-PM and MLR methods. Chang et al. (2010) used

different methods to estimate pan evaporation, including also the KSOM and the FAO56-PM. According to the results of Chang et al. (2010), K-SOM was the best of the studied methods, and it was found that the Penman-Monteith method is also likely to underestimate evaporation. Malik et al. (2017) used four heuristic approaches and two climate-based models to approximate monthly pan evaporation, where the K-SOM model performed better than the climate-based models. The regression line in scatter plots has R2 as 0.937 for K-SOM model at Pantnagar and Ranichauri (India), respectively. In the study of Malik et al. (2017), RMSE values were 0.685 and 1.126 for K-SOM, when 50% of the total available data was used in the testing of models in two stations. This section should be put in the introduction.

Line 280 The regression line in scatter plots has R2 as 0.937 for K-SOM model at Pantnagar and Ranichauri (India), respectively. 'Respectively' can be deleted.

Can the confidence of the correlation coefficient pass the significance test?

---

## Author Comment (AC1)

**RC1**

The manuscript describes an interesting phenomenon -- but doesn't explore plausible explanations. One may expect that the water temperature in the Class A pan is influenced by sediment at the bottom of the pan, and by having waterplants that limit water circulation. The temperature at the water surface will influence evaporation. Even measurements of surface water temperature for a relatively short period could help quantify such effects. One expects the temperature differential to be highest under full-sun conditions and lowest with an overcast sky. So plotting the temperature differential to environmental conditions could give some indication of the mechanism involved.

The statistical toolbox used is rich -- but one wonders how replicable results might be under conditions beyond those of the experiment if there is no mechanistic understanding of the process. The 'machine learning' methods are deemed successful in 'fitting', but results are not presented in a way that allows others to use them in new settings.

Dear Reviewer,

thank you very much for your constructive comments.

The macrophytes seeded in the pans probably hinder the water circulation and the radiation penetration into the deeper water layers. This is the reason why the surface water temperature in seeded pans is higher than in empty A pan. This summer, water temperature layering was also detected in A pan. We plan to complete the manuscript with this new result. Comparison in temperature layering between seeded and empty pans highlights the macrophyte modified circulation patter.

Surface water temperature ($T_w$) of A pans was sensed with thermocouples (Delta Ohm HD-226-1) at 10-min intervals in 2015-2020.

[Figure]

Figure. Water surface temperature ($T_w$) of different pan treatments (C – control, S – Class A pan with sediment cover bottom, SM – Class A pan with submerged macrophyte) in sample days: a), b) and c) days with clear sky; d), e) and f) days with overcast sky.

Differences detected in the $E_p$ of A pan treatments could be explained by differences in $T_w$. Figure 5 illustrates the change in $T_w$ within a day, on rainless days. Figures 5a-c show the $T_w$ values measured on full-sun days in hourly terms. On clear days, $T_w$ of C was lower from sunrise to sunset, compared to $T_w$ S and SM. In summer, on full-sun days, the daily mean $T_w$ of the water with submerged freshwater macrophytes was 2.73-4.09 and 3.38-4.82% higher, compared to $T_w$ of C and S, respectively. The $T_w$ of S was 0.68-1.25% higher compared to $T_w$ of C.

The differences in $T_w$ on overcast sky days are very small for A pan treatments (Fig. 5d-f), compared to clear sky days. These days $T_w$ of C was 1.70-2.10 and 0.89-1.25% lower than $T_w$ of S and SM, respectively. Based on the measurements, the $T_w$ of S appeared higher even compared to $T_w$ of SM, although the difference was very small (0.55-0.82%).

Higher evaporation measured of S and SM could be due to higher $T_w$. In the sheltered, shallow water bodies, the vegetation alters the radiation properties within the water, as a result in reducing stratification (Coates and Folkard, 2009). Furthermore, the submerged plants in water have obvious influences in modifying the water transparency, the heat transfer inside the water (Golosov and Kirillin, 2010) and eddy motion (Wang et al., 2019). The reason for the higher $T_w$ above submerged macrophytes could be the more heat absorbed at the upper water column, while the bottom temperature falls (Sharip et al., 2012) than that of in comparison to empty pans. In the case of natural waters containing submerged macrophytes, the plants restrain the radiation transfer to the deep water and reduce it to 0 at a depth of 1 m (Wang et al., 2019).

Coates, M.J., Folkard, A. (2009): The effects of littoral zone vegetation on turbulent mixing in lakes. Ecol. Model., 220, 2714–2726.

Golosov, S.; Kirillin, G.A. parameterized model of heat storage by lake sediments. Environ. Model. Softw. 2010, 25, 793–801.

Sharip, Z., Hipsey, M.R., Schooler, S.S., Hobbs, R.J. (2012): Physical circulation and spatial exchange dynamics in a shallow floodplain wetland. Int. J. of Design & Nature and Ecodynamics. 7,3, 274–291.

Wang, Y., Ma, Q., Gao, Y., Hao, X., Liu, S. (2019): Simulation of the Surface Energy Flux and Thermal Stratification of Lake Taihu with Three 1-D Models. Water, 11(5), 1026.

Sincerely,

Brigitta Simon-Gáspár

---

## Author Comment (AC2)

Dear Reviewer,

we thank the reviewer for appreciation of work and for the comments. We believe the current comments can greatly help improve the quality of the paper. Please find our responses in the attached file.

General comments

The authors used FAO-56 Penman-Monteith equation (FAO56-PM), multiple stepwise regression (MLR) and Kohonen self-organizing map (K-SOM) techniques to estimate daily pan evaporation (Ep) in three treatments. And in an six-season experiment. The 10 modelling approach included six measured meteorological variables were compared and evaluated. The results showed that the MLR method provided close compliance with the observed pan evaporation values, but the K-SOM method gave better estimates than the other methods.

I really appreciate the huge work made by the authors. However, I have some questions about the innovation of the research method and the purpose of this study. First, the ,methods of FAO-56 Penman-Monteith equation (FAO56-PM) (Allen et al., 1998), Kohonen self-organizing map (K-SOM) techniques (Kohonen, 1982) and multiple stepwise regression (MLR) are classic, but also relatively old methods. Predecessors have done a lot of research and published a lot of relevant papers on the comparison of these methods. If only a few traditional, classical methods are compared and discussed, I think that there is not enough innovation in terms of methodology to be published in HESS, and there are already many ready-made papers on comparative studies of evaporation calculation methods. In addition, the authors said that "there is little information in the literature on how submersed macrophytes affect the evaporation of a lake" in Introduction (Lines 64). Is this statement supported by the literature? (Wang, J.H. 1994. Effects of aquatic plants on water surface temperature and evaporation. Arid Land Geography, 17(2), 3. doi: CNKI:SUN:GHDL.0.1994-02-009). This article is not available in any traditional datnbase (scimago, google scholar, Sci-hub. se). Only the Journal name was found here: http://english.egi.cas.cn/pub/alg/. But even in the search window, the article is missing.

The aim of the study was to investigate the effect of littoral sediment and macrophytes on lake evaporation and not an introduction of a new method in pan evaporation estimation. The previous results in FAO-56 Penman-Monteith equation (Allen et al., 1998), Kohonen self-organizing map techniques (Kohonen, 1982) and multiple stepwise regression are classic methods, highlighted widely by citations in the study. They are the tools in analysing the effect of sediment and macrophytes in pan (lake) evaporation estimation only. The novelty of the paper is the way how the evaporation estimation is carried out.

The second main aim of this paper was to estimate daily Ep using FAO-56 Penman-Monteith (FAO56-PM), Kohonen self-70 organization map (K-SOM) and multiple stepwise linear regression (MLR) methods. Since this purpose is only the comparison and evaluation of several traditional methods, and I personally still feel that the innovation is not sufficient for HESS.

This work's novelty is not the use of a new statistical method. We complete the new aim of the article as follows: "Up to our best knowledge, there are no studies attempting to project the lakes evaporation using traditional A pan measurements, taking the macrophytes- and sediment-related factors into account. Lake evaporation study is missing under such climate conditions as our experimental site."

Specific Comments

1. English language needs to be modified. I found several unclear sentences that make it difficult to understand the analysis and results carried out.

Thank you for your comment. Although, this manuscript was corrected by an English native speaker. Correcting the paper, we repeat this process. It would be useful to know which are those unclear sentences.

2. The description at the beginning of the Abstract is too simple and empty, two to three sentences should be used to focus on the shortcomings of the current study and the innovation of this study.

The Abstract will be completed to highlight the main novelty and the mechanical impact of seeded macrophytes on the pan evaporation.

3. Some of the references in the Introduction are too old. It is suggested that the author update some relevant studies recently published.

We added 25 new references to the Introduction.

4. The font resolution in Figure 1 is too low to see the relevant text clearly. I suggest the author to redraw it.

Done.

5. The numeric font in equation 2 is suggested to be Times New Roman, and the rest of the formula is the same.

Done. Thank you for your comment.

6. In Materials and methods, it would be better to give specific steps about the experimental design of this study, the current presentation is relatively sketchy.

We complete the materials and methods with more exact steps of the experimental design as follows:

"The used sediment was psammal/psammopelal (Ø > 6 μm - 2 mm, sand/sand with mud) with the following composition: quartz, calcite, aragonite, dolomite, muscovite, chlorite, feldspar, smectite, kaolinite and pyrite (Anda et al., 2016). One Class A pan was treated with a 0.02 m thick sediment–covered bottom.

Macrophyte samples were gathered from lake Balaton (Keszthely Bay) with similar water depth (0.6–0.8 m) each year. The amount of crop density was controlled monthly without variation in the green mass weight of crops between natural habitat and "seeded" US Class A pans. In the experimental area three species of submerged, freshwater aquatic macrophytes: Potamogeton perfoliatus, Myriophyllum spicatum and Najas marina was colonized."

7. The Results and Discussion session. I found some good results from this study, but unfortunately the authors' description of these results is too brief (Both in Figures and Tables)and suggest a more specific analysis and evaluation of the results. And the discussion was not in-depth enough and was only a brief description of the Results. It is recommended to fully evaluate and discuss the results obtained by several different methods used in this study in terms of the mechanism of influence.

The results will be analysed in more detail accounting the following:

We plan to extend the description of the results with weather conditions of the studied six seasons using Thornthwaite index, TI (). In the next step, the evaporation will also be analysed based on TI. We discuss differences in pan evaporation based on actual weather conditions.

A more specific analysis and evaluation of the missing discussion of pan evaporation related to thermal layering and surface water temperature will also be added. These basic observations in terms of the mechanism of influence will be completed to Abstract as well.

8. The content of the conclusion should not be a simple retelling of the results and discussion, but also a more in-depth explanation of the scientific significance and potential application value of the study, rather than the kind of formulaic statement in the last paragraph of the conclusion.

Potential application value of the study is validating presence of littoral sediments and macrophytes in evaporation estimation, the amount of lost water by lakes that can easily be accounted in the prediction of wetland performance. Results from the study may also be contributed to the protection of aquatic plants and environmental management of lakes in other regions of the world. Management strategies aiming to estimate accurate water budget terms including evaporation can be a realistic aim for preventing further inaccurate water loss projections.

9. I am not quite sure if the current hess format requires line numbers to be marked every five lines, which causes some reading difficulties, if not required by the journal format, it is recommended that authors mark all line numbers.

The authors used the Copernicus_word_template format recommended by HESS.

Sincerely,

Brigitta Simon-Gáspár

---

## Author Comment (AC3)

**RC3**

We thank the reviewer for appreciation of work and for the comments. We believe the current comments can greatly help improve the quality of the paper. Please find our responses in the attached file.

This paper discusses a way of modeling pan evaporation using 3 methods: the Penman-Monteith equation, multiple step-wise regression, and the Kohonen self-organising map. The novel element to this work appears to be the fact that the pan evaporation was measured with 3 class A pans but two contained sediment and one contained submerged macrophytes on top of sediment. To me, this is the part that sets the paper apart and is important, because the authors are trying to create more realistic conditions for the observational experiment before using the data for a modelling study.

Thank you for your understanding!

I liked the part of the discussion lines 248-250 where the authors discuss the low winds in this study and in earlier studies. I think that the fact that they honestly state a kind of negative result is helpful and proper.

Thanks again!

1. I was disappointed that the methods were not discussed in more detail, in particular, the Kohonen self-organising map method. I can see that proper citations are given, and that this method has been applied to evaporation modelling before. However, I think that the paper would benefit greatly from an introduction to the Kohonen method. Then all 3 methods could be explained to the reader alongside the relevance to the physical process being studied. Why were these methods chosen in the first place? What are the pros and cons to these methods?

K-SOM provides an indirect method for the estimation of pan evaporation, that seems necessary to get evaporation of natural ecosystems including lakes. In other words, applying an indirect method, in which the pan evaporation is estimated from other, easily measurable meteorological parameters such as sun radiation, air temperature and relative humidity has of primary importance. This approach has widely been used for pan evaporation projection among others by Kisi et al. (2016) and Lin et al. (2013). Kisi et al (2016) compared the soft computing model K-SOM and multiple linear regression (MLR). The authors demonstrated the superiority of K-SOM over MLR even in the model performance.

The Penman-Monteith model is considered as the international standard for computing potential evapotranspiration and predicting crop water requirement. Penman-Monteith equation (FAO-56) may also be proper method to get pan evaporation with submerged macrophytes. Wang et al. (2021) reported that actual evaporation is important for hydrological research due to its direct impact on the hydrologic processes (water cycle, water resources management). The above authors concluded that to estimate pan evaporation, it is essential to find the proper formulation of Penman-Monteith equation. It may be especially true even in pans with seeded macrophytes. In accordance with composition of lake ecosystems, this is the method in evaporation estimation that implies living organism.

To our best knowledge, no similar work has been published previously using the three modelling methods for seeded pan evaporation estimation. (This last sentence goes to Introduction)

Added references:

Kisi, O., Genc, O., Dinc, S., Zounemat-Kermani, M.: Daily pan evaporation modeling using chi-squared automatic interactiondetector, neural networks, classification and regression tree, Comput. Electron. Agr., 122, 112-117, DOI: 10.1016/j.compag.2016.01.026, 2016.

Lin, G.F., H.Y. Lin and M.C. Wu Development of a support-vector-machine-based model for daily pan evaporation estimation. Hydrol. Process., 27 (22) (2013), 3115-3127

Wang, L., Hang, S., Tian, F., Comparison of formulating apparent potential evaporation with pan measurements and Penman methods, J Hydrology, 592, January 2021, 1258162021, https://doi.org/10.1016/j.jhydrol.2020.125816

2. Figure 5 should be explained. I can see that this is related to figure 3 but the links are not made clear. The x and y axes are not labelled in either figure. How do the hexagons map to the inputs or outputs? I can't figure this out from reading the manuscript. Also, I found no definition for "importance" in the caption. This is part of the lack of time spent discussing and explaining the relevance of the methods used in the study.

These are not traditional Fig. with x and y axis. The Fig. 3 is a schematic layout of correlations, the winning nodes, and their neighbourhood.

To Fig. 3: "As similar input patterns could have different outputs, to determine the best output for a given input pattern is to use the mean output value as the clustered input patterns to the correspondent neuron, and then the closest (most similar) neuron would be directly used for the given input pattern (Chang et al., 2010, Kohonen, 1990)."

Description of Fig. 5 can be found in lines 215-220.To Fig. 5: "Superimposed on K-SOM patterns of input meteorological variables, radiation, air temperatures including minimum and maximum, relative humidity, wind speed could be captured revealing their co-variability with the pan evaporation."

To References

Kohonen, T., 1990. The self-organizing map. Proceedings of the IEEE 78 (9), 1464–

1480.

F-J. Chang, L-C. Chang, H-S. Kao, G.-R. Wu, Assessing the effort of meteorological variables for evaporation estimation by self-organizing map neural network. Journal of Hydrology, 384, 1–2, 118-129, 2010, https://doi.org/10.1016/j.jhydrol.2010.01.016

3. Tables 1 and 2 are very large and comprehensive. While I think that the information is critical to explaining the conclusions of the paper, I do not think that the information in them is easy to make sense of. Is there any way that the tables could be re-organised or even some of the information could be turned into figures to display the information in a clearer way? One suggestion for table 2 is to keep  information for the correlation values only in the tables, but to put all the statistics (max, min, mean, std. dev.) in a figure with sub panels.  Are all the statistics relevant?  Perhaps the authors could be a bit more selective?  I agree that information on the full, training and testing data should be presented.

The Table 1 is important. We keep it in its original form. In case of Table 2 some elements will be omitted (CV, max and min).

4. Figure 1 was not displayed with very high resolution. Could the authors provide a higher quality figure?

Yes

5. Figure 4 box plot whiskers and circles are not clearly defined. Software packages which compute these types of diagrams are not all the same. Please could the range (and meaning) of the box lenghth, whiskers and circles be stated explicitly.

Completed the title of the Fig: "The lower and upper ends of the box indicate the 25th and 75th percentiles of the variances, respectively, while the horizontal bar within the box indicates the median. The two horizontal bars indicate the range that covers 90% of the variances. Outliers are indicated with circles."

6. Table 1 has three lines at the bottom saying "Based on observed means" but I don't know what these lines are referring to. Is it the full, training and testing data sets?

We correct it

Minor corrections

In figures with multiple panels, the sub-panels should be labeled with letters (a, b, c,...) in order to make the discussion of the results clearer in the body of the paper and make it easier to clarify the definitions in the captions.

We clarify where it is necessary

L167 "displays a regular pattern" does not make sense to me. Can the authors make clearer what the Relative humidity and global radiation statistics are exhibiting and why it is important to notice this?

The last two sentences will be omitted from the manuscript.

We plan to extend the description of the results with weather conditions of the studied six seasons using Thornthwaite index, TI. In the next step, the evaporation will also be analysed based on TI. We discuss differences in pan evaporation based on actual weather conditions. See also answers of Ref 2.

L178 Where is F defined? I could not find it.

Completed

L241 Please change "few" to "little"

Done

L247 Please change "researches" to studies"

Done

L273 Please change neural network to "a neural network approach".

Done

L290 the range or tuple presented is unclear. Is "(-0.42-0.44)" really (-0.42 to -0.44) or (-0.42, -0.44)?  Also, it seems strange to put the larger number -0.42 in front of -0.44 if indeed they both are negative.

Clarified

L296 Please change "has high priority" to "is superior to".  I'm not sure about the phrase "prediction precision"  precision sounds like a computational error here. Are you talking about skill? High correlation or low RMSE?  Is there a better way to phrase this?

Modified as accuracy in prediction

---

## Author Comment (AC4)

**RC4**

**Reviewers' comments:**

In this paper, K-SOM and other methods are used to improve the estimation of lake evaporation, which is conducive to accurately estimate the total lake evaporation and improve the climate effect of the lake under the background of climate change. I recommend publication of the paper in HESS after revision.

**Thank you!**

**Major comments**

Has this article been studied by simulation at Keszthely, Hungary. However, how do you consider the effect of the evaporation of non-uniform underlying surfaces, such as mountains, and grass?

The use of class A pan to measure evaporation is controlled by WMO Guide including its placement, pan dimensions, time, and frequency of measurements etc. The aim of these strict restrictions is the comparability of evaporation measurements under different environmental conditions. The goal of the study was to detect the effect of littoral sediment and macrophyte on evaporation rate. The seeded and empty pans were set up on the same place on the Agromet. Research Station of Keszthely, allowing comparison of standard and seeded pan's evaporation rates. Finally, the impact of macrophytes and sediments on (lake) evaporation rates were discussed.

This paper improves the calculation method of lake evaporation, and further analysis of lake evaporation and its climate effects are needed on the Lake Balaton in the future.

**Thank you!**

**Minor comments**

Figure 1 should be topography.

**Corrected**

L.47 The unsupervised NNs, including Kohonen Self Organizing Maps (K-SOM), has several advantages (Kohonen, 1982). Full name should be given for the first occurrence'NN'.

**Completed**

L.16-20 Performances of the different models were compared using statistical indices, which included the root mean square error (RMSE), mean absolute error (MAE), scatter index (SI) and Nash-Sutcliffe efficiency (NSE). The results showed that the MLR method provided close compliance with the observed pan evaporation values, but the K-SOM method gave better estimates than the other methods. Overall, K-SOM has high accuracy and huge potential for Ep estimation for water bodies 20 where freshwater submerged macrophytes are present. This section need to be rewrite.

Performances of the different models were compared using statistical indices. The results showed that the MLR method provided close compliance ( $R^2$ =0.58-0.62) with the observed pan evaporation values, but the K-SOM method ( $R^2$ =0.97-0.98) gave better estimates than the other methods. Although, we plan to re-write the whole Abstract.

L.84 (latitude: 46°44′N, longitude: 17°14ʹE, elevation: 124 m above sea level) 'above sea level' Can be abbreviated as a.s.l.

**Done**

L.231 From the figure, it can be observed that most of the estimated daily Ep values are close to the observed daily Ep values for all three pan treatments.

Which figure?

**In Fig. 6. We complete the text.**

Many researchers have conducted research with neural networks aimed at the estimation of Ep as a function of meteorological variables (Keskin and Terzi, 2006). Several of these researchers found better results in Ep estimation with neural network than those obtained from the Priestley-Taylor and the Penman methods (Rahimikhoob, 2009; Malik et al., 2020). Consistent with other studies, this study demonstrated that modelling of Ep is possible through the use of K-SOM technique in addition to the 275 FAO56-PM and MLR methods. The comparison results indicated that, in general, the K-SOM model was superior to the FAO56-PM and MLR methods. Chang et al. (2010) used different methods to estimate pan evaporation, including also the KSOM and the FAO56-PM. According to the results of Chang et al. (2010), K-SOM was the best of the studied methods, and it was found that the Penman-Monteith method is also likely to underestimate evaporation. Malik et al. (2017) used four heuristic approaches and two climate-based models to approximate monthly pan evaporation, where the K-SOM model performed better than the climate-based models. The regression line in scatter plots has R2 as 0.937 for K-SOM model at Pantnagar and Ranichauri (India), respectively. In the study of Malik et al. (2017), RMSE values were 0.685 and 1.126 for K-SOM, when 50% of the total available data was used in the testing of models in two stations. This section should be put in the introduction.

**We don't think so. This paragraph contains our results in the light of other author's ones dealing with the same topic as this study.**

Line 280 The regression line in scatter plots has R2 as 0.937 for K-SOM model at Pantnagar and Ranichauri (India), respectively. 'Respectively' can be deleted.

**Deleted**

Can the confidence of the correlation coefficient pass the significance test?

Yes, all p levels are below 0.001. We indicate them on Fig. 6 as well.

---

## Author Comment (AC5)

RC5

In this manuscript, the authors use three methods to calculate daily evaporation during summer months (Jun to Sep) at a experimental site in Hungary. The three methods are the FAO Penman Monteith (PM) method, multiple stepwise regression and Kohonen self-organizing (SOM) maps. All methods are provided with observed meteorological time series for the years 2015 to 2020 as input. Simulated values are then compared to the observed evaporation estimates of three evaporation pans. The difference between the three pans is that one is a standard open water pan, one pan is partly filled with sedimentes and one with water living plants. Overall, the Kohonen self-organizing maps yield by far the closest match to observed evaporation estimates for all three pans.

The topic of evaporation estimates is fundamental to land-surface hydrology. For example, the PM method is frequently used to estimated potential evaporatin in hydrologic models. Errors in the PM methods will bias any subsequent hydrologic modeling. Despite the fact, that the topic of the manuscript is of general importance, the structure of the manuscript is poor and needs to increase substantially. One important issue is that I could not find that all findings listed in the abstract and the conclusions mentioned in the abstract are supported by the main text. Additionally, the methods part does not provide all details to follow what the authors did exactly. This does concern the PM method and the SOM method. The re-phrasing of the mentioned sections is in process and see also answers of RC2 and RC3. To complete the M&M section based on the answer of RC3 is as follows: "K-SOM provides an indirect method for the estimation of pan evaporation, that seems necessary to get evaporation of natural ecosystems including lakes. In other words, applying an indirect method, in which the pan evaporation is estimated from other, easily measurable meteorological parameters such as sun radiation, air temperature and relative humidity has of primary importance. This approach has widely been used for pan evaporation projection among others by Kisi et al. (2016) and Lin et al. (2013). Kisi et al (2016) compared the soft computing model K-SOM and multiple linear regression (MLR). The authors demonstrated the superiority of K-SOM over MLR even in the model performance.

The Penman-Monteith model is considered as the international standard for computing potential evapotranspiration and predicting crop water requirement. Penman-Monteith equation (FAO-56) may also be proper method to get pan evaporation with submerged macrophytes. Wang et al. (2021) reported that actual evaporation is important for hydrological research due to its direct impact on the hydrologic processes (water cycle, water resources management). The above authors concluded that to estimate pan evaporation, it is essential to find the proper formulation of Penman-Monteith equation. It may be especially true even in pans with seeded macrophytes. In accordance with composition of lake ecosystems, this is the method in evaporation estimation that implies living organism."

"To our best knowledge, no similar work has been published previously using the three modelling methods for seeded pan evaporation estimation." (This last sentence goes to Introduction)

Added references:

Kisi, O., Genc, O., Dinc, S., Zounemat-Kermani, M.: Daily pan evaporation modeling using chi-squared automatic interactiondetector, neural networks, classification and regression tree, Comput. Electron. Agr., 122, 112-117, DOI: 10.1016/j.compag.2016.01.026, 2016.

Lin, G.F., H.Y. Lin and M.C. Wu Development of a support-vector-machine-based model for daily pan evaporation estimation. Hydrol. Process., 27 (22) (2013), 3115-3127
I am convinced that the comparison of evaporation estimation methods for the three pans (in particular Figure 6) are valuable findings that should be reported, but the manuscript needs to be

substantially improved. Thank you!

Most importantly, the authors should decide what their main focus of the study is. Is it the effect of macrophytes on evaporation or is it the different estimation methods? At the moment, this is not clear. I would suggest the former to be the more interesting subject.

The content of the study aim is also under consideration. But the macrophyte's effect in pan evaporation is in the focus. We complete the new aim of the article as follows: "The aim of the study was to investigate the effect of littoral sediment and macrophytes on lake evaporation and not an introduction of a new method in pan evaporation estimation. The previous results in FAO-56 Penman-Monteith equation (Allen et al., 1998), Kohonen self-organizing map techniques (Kohonen, 1982) and multiple stepwise regression are classic methods, highlighted widely by citations in the study. They are the tools in analysing the effect of sediment and macrophytes in pan (lake) evaporation estimation only. The novelty of the paper is the way how the evaporation estimation is carried out. Up to our best knowledge, there are no studies attempting to project the lakes evaporation using traditional A pan measurements, taking the macrophytes- and sediment-related factors into account. Lake evaporation study is missing under such climate conditions as our experimental site."

Abstract:

L. 14ff: I don't think that the statement regarding the correlation of RH is supported by the findings of this study. See my comment below regarding L. 218ff.

We omit the word "stronger"

The conclusion mentioned in the abstract on line 19f is not given anywhere else in the manuscript. It is unclear how the authors come to this conclusion or what they mean with "potential".
We mean potential as possibility. The word potential will be changed in possibility

Introduction:

Section starting at line 52 is a collection of statements that do not follow a apparent logical structure. It is not clear to me what the authors wish to express here.

Re-written Introduction part is: "Evaporation of open water surfaces is usually measured by means of pans endowed with unrealistic properties. These pans are filled with clean tap water and the evaporated water is also replaced with tap water that is not the case in natural ecosystems. In nature, however, there may also be submerged macrophyte living in the open water. These plants presence is essential and affects the chemical and physical water properties including its quality (Yan et al., 2019). Furthermore, the species that are rooted in the sediment can stabilize the sediment by inhibiting its resuspension (Madsen and Cedergreen, 2002; Vymazal, 2013).
Changes in the heat regime of a water body had been reported to result in alterations of macrophyte community composition (Barko et al., 1982), which may affect the temporal appearance and spatial distribution of macrophytes in the future. As a result, due to global climate change, it is important to examine submerged macrophytes in all aspects, including their effect on evaporation. "

Methods:

L. 110f: The data described in Section 2.1 is not sufficient to apply the Penman-Monteith equation. Section 2.1 states that global radiation $R_s$ is measured but Penman-Monteith equation requires net radiation and ground heat flux. How are the latter two derived?

We complete the M&M: "Rn was estimated from global radiation, mean daily temperature, the mean daily vapor pressure, the site latitude and elevation after Allen et al. (2005). A fixed value of 0.23 was applied for the common reed albedo. It was assumed that soil heat flux density was G= 0 on a daily basis. Detailed description of the process can be read in Soós and Anda (2014)."

References:
Allen, R.G., Clemmens, A.J., Burt, C.M., Solomon, K., O'Halloran, T., 2005. Prediction accuracy for projectwide evapotranspiration using crop coefficients and reference evapotranspiration. J. Irrig. Drain. Eng. ASCE 131 (1), 24–36
Anda, A., G. Soos, J. A. Teixeira da Silva, V. Kozma-Bognár 2015. Regional evapotranspiration from a wetland in Central Europe, in a 16-year period without human intervention, Agric. Forest Meteor. 205: 60-72, DOI: 10.1016/j.agrformet.2015.02.010
Soós, G., Anda, A, A methodological study on local application of the FAO-56 Penman-Monteith reference evapotranspiration equation, GEORGIKON FOR AGRICULTURE: A MULTIDISCIPLINARY JOURNAL IN AGRICULTURAL SCIENCES 18, 2, 71-85, 2014.

Results:

L. 187f: There are only four lines describing the results of table 2. This is not well balanced. Either the text needs to be expanded or the table shortened.

To the text completed with: "There was statistically significant difference in evaporation rates of full datasets as well as in case of training and testing datasets between the seeded and classic Class A pan."
Table 2 will also be re-structured by omitting some of the lines (CV, max and min)."

L. 206f: I am not an expert in self organizing maps. I don't know how to interprete characteristics shown in table 3 and the authors only describe the last two lines in this table.

We add the following to Table 3 explanation: "These are the usual parameter table for K-SOM. The inputs were normalized, the code book is generated, the map size is the dimensions of the component planes (Figure 5), the neighbouring function of the pixels is Gaussian, the shapes of component planes are sheets, the lattices in planes are hexagonal."

L. 218ff: "Thus, the correlation..." This sentence is confusing to me. First, it should state observed values and not modeled values. Second, it is shown in table 2 that RH is negatively correlated with E_p which is expected. Here, the authors state that red colors in Figure 5 show high correlation. For RH, the values are substantially higher than for any other variable suggesting a higher impact. This suggests to me that the SOM algorithm is not able to reproduce the relationships reported in table 2.

The Fig. 5 is in accordance with results in Table 2. In the two component planes of Fig. 5 (Ep and RH), the red areas of hexagons are on the opposite sides that indicate the negative correlation as it was stated in L218ff. That is in the text: "The component planes also visually confirm the negative correlation between RH and Ep, with high values of the RH resulting in low values of the Ep"

L. 231: I disagree with this statement. How can the authors state that all three methods are close to observed values, when coefficient of determination varies from 0.11 for Penman-Monteith method to 0.97 for self-organizing maps. I think it is fair to state that the Penman-Monteith method is not

able to reproduce the observed values. It is not clear to me whether the authors did apply the Penman-Monteith equation correctly because not all details are provided in the manuscript (see comment above).

This sentence will be omitted. The asked information was completed in M&M. See above

Discussion:

L. 256f: There are results reported in the discussion section. This should be moved to the results section and is not a clear manuscript structure.

We move L 256f to the results

L. 271ff: This section lists findings of other studies but does not provide a discussion of these results against the findings of the present study.

It demonstrates previous results that existed before this study. The conclusion contains the new findings of this study.

Minor comments:

L. 9: There is a misleading typo here: the A should not be capital.

Thank you!

L. 110f: Which equation was used to derive e_s and e_a from RH?

The Tetens equation (Monteith and Unsworth, 2008; Allen et al., 1998; Tetens, 1930) was used for calculating saturation vapor pressure ($e_s$) as follows:

$e_s=0.6108*\exp(17.27T_a/(T_a+237.3))$, where $T_a$ is the air temperature in ºC. The vapor pressure, $e_a$ was calculated from the relative humidity (RH):

$e_a=(RH/100)*e_s$

References:
Allen RG, Pereira LS, Raes D, Smith M. Crop Evapotranspiration-Guidelines for Computing Crop Water Requirement, Rome, Food and Agriculture Organization of the United Nations, 1998.
Tetens O. Über einige meteorologische Begriffe. Z. Geophys., 1930; 6. 297-309.
Monteith, J.L., and Unsworth, M.H. 2008. *Principles of Environmental Physics*. Third Ed. AP, Amsterdam.

L. 118ff: Section 2.4 is not understandable to readers who are not familiar to SOM. It needs to be rewritten using an easier language. Figure 3 is also very confusing. Also, E_p is mentioned in Figure 3 as input variable, but this cannot be correct. I guess that observed E_p is used during training to compute an error measure.

According to answer RC3, the description of K-SOM will be extended.
Thank you for your suggestion, the observed Ep was used during the training. The Fig. 3 is only schematic representation of K-SOM network architecture. We correct the input of Fig. 3 by omitting the Ep.

L. 139: The sentence regarding the splitting of the data is incomplete.

We complete the sentence

L. 184: Table 1 can be moved to the appendix because it is not central to the goal of the manuscript.

Thank you!

---

## Author Comment (AC6)

RC6

The comment "Most importantly, the authors should decide what their main focus of the study is. Is it the effect of macrophytes on evaporation or is it the different estimation methods? At the moment, this is not clear. I would suggest the former to be the more interesting subject." gives relevant advice to the authors.

In revising the manuscript, authors may challenge the concept that there is a single 'potential evaporation' metric that applies to all vegetation or open-water surfaces: the presence of water plants in Class A pans will influence surface temperature (hopefully you have some data on this) and hence evaporation. Clarifying the physical basis of this effect will be more relevant for future authors than a comparison of 'interpolation techniques' within the existing data set (with little confidence in using results elsewhere.

According to the RC-6 reviewer's advice (this is his second term), the clarified physical basis of the pan with submerged macrophyte was added to the manuscript. See below. At the same time, we omit the Fig. provided in the answer of RC1. Instead, the analysis below completes the manuscript.

The aim of the study will be clarified as previous answers present it.

To M & M

In the last vegetation period, to detect vertical $T_w$ profiles, four fastened thermistors of Delta Ohm HD-226-1 (accuracy: 0.3ºC) collected the temperature data at 0.05, 0.10, 0.15 m depth from the pan bottom and on the water surface, at 10-min intervals. Hourly averaged $T_w$ values were used in the analysis. To present diurnal variation in $T_w$ and stratification, sample days were selected for clear-sky, calm, and cloudy weather conditions.

To Results

On the basis of daily variation of $T_w$ in different depth, two time-periods were distinguished (Fig. 1); daytime (7:00 – 18:00 h, LMT) and nighttime cooling (19:00 – 6:00 h, LMT).

[Figure]

Fig. 1 Water temperature of different pan treatments (C – Class A pan/control; S – Class A pan with sediment covered bottom; SM – Class A pan with submerged macrophyte) in clearsky and cloudy sample days

On clear sky conditions, the surface $T_w$ peaked at 14:00 h, irrespective to treatment. The magnitudes of surface $T_w$ in daytime (between 07:00 and 14:00 hours) increased from 21.6 to 37.5⁰C in C, from 23.0 to 37.4⁰C in S, and from 19.8 to 38.0⁰C in SM. Then, with declining solar radiation, the $T_w$ slightly decreased during the nighttime cooling to 21.2, 21.8 and 18.7⁰C in C, S and SM, respectively, until sunrise. In deeper water depth, a similar pattern of $T_w$ with slightly smaller magnitudes was measured with time lag of 1-to-2-hours from the surface $T_w$. In classic A pan, the $T_w$ in deeper depth from the surface did not reduce as rapidly as $T_w$ in seeded pans. On cloudy days, insignificant $T_w$ differences less than 1⁰C (p=0.059 - 0.969) between the neighbouring layers were observed in every treatment.

To Discussion

Daily mean $T_w$ increases were 5.4 and 4.5⁰C in S and SM, respectively, compared to C during clear-sky conditions. Despite the less intense stratification on overcast days, $T_w$ of seeded pans was 5.4⁰C higher than that of daily mean $T_w$ of C.

Increased stratification was evident in daytime, but the number of layers strongly depended on macrophyte presence. More moderate $T_w$ layer differences were also present at night. The stratification was the most intense with 3 significantly different layers (p<0.001) in seeded pans, during clear-sky daytime. At the same time, the number of layers with varied $T_w$ was only 2 (p<0.001 – p=0.012) in classic A and sediment covered pans. Results in the study were confirmed by Andersen et al. (2017) concluding that shallow lakes colonized by submerged macrophytes strongly stratify the water body, mainly during the daytime. The reason of this stratification is the dissipating turbulent kinetic energy and absorbing heat (Vilas et al., 2018). The plants may act as a barrier to seeded pans water mixing, attenuating underwater light, thereby enhancing the thermal stratification inside the pan's water column.

The strength of stratification, the daily mean $T_w$ differences between the surface and bottom water were 2.5 (p=0.005), 3.0 (p<0.001) and 6.5⁰C (p<0.001) in C, S and SM, respectively, on cloudless days. At night-time cooling, variation in $T_w$ between different layers was less pronounced, remaining below 1⁰C (p<0.001 – p= 0.005).

In addition to stratification, the macrophytes have strengthened the daily variation of $T_w$ in different depth. A 0.3⁰C increase in daily mean surface $T_w$ of seeded pans related to C was obtained during daytime, with variation ($T_{max} – T_{min}$) of 18.4 and 19.3⁰C in C and SM, respectively. On the bottom, an opposite trend in daytime mean $T_w$ was detected; the seeded pans $T_w$ in 0.05 m depth was 3.1⁰C (p=0.040) cooler than that of the $T_w$ of C. Probably the macrophyte presence resulted in insufficient downward heat transport, maintaining the more stratified water body of seeded pans. Herb and Stefan (2004) also found reduced turbulent mixing in shallow Otter Lake, Minnesota, with rooted macrophytes. The authors observed that $T_w$ fluctuations at 20 cm depth were 3⁰C in open water and 4.5⁰C in lake water with macrophyte cover. Evapotranspiration functions of SM fitted to surface $T_w$ evolution; the higher the surface $T_w$, the more intense the $E_p$ rate was measured in SM related to $E_p$ of classic A pan.

To Conclusion

Macrophyte induced thermal stratification in water bodies (lakes/evaporation pans) emerge only in the vegetation period, during macrophytes development. One less layer in Classic A pan compared to macrophyte seeded pans was probably due to modified $T_w$ stratification causing changed water column stability. Wider $T_w$ values induced dynamics presented in the macrophyte seeded pans demonstrated the possibility of developing a more heterogenous environment for aquatic ecosystems. Macrophyte induced modified thermal stratification with higher surface $T_w$ could explain the increased $E_p$ in seeded pans. Modified $E_p$ of seeded pan made those values closer to the $E_p$ of natural lakes with submerged macrophytes. While $T_w$ stratification trend in SM was similar to that of natural shallow lake's one, it may also provide a new consideration for routine hydrometeorological management. $T_w$ distribution in macrophyte covered lakes impacts other physical properties such as nutrient cycling, dissolved oxygen etc. When treating $E_p$ from a pan to that from a vegetated surface including lakes or other aquatic habitats, to improve evaporation estimation, multidimensional approximation is necessary offering simple methods for end-users including hydrologists, meteorologists, or any other specialists.

To References

Vilas M.P., C. L. Marti, C. E. Oldham, M. R. Hipsey, 2018. Macrophyte-induced thermal stratification in a shallow urban lake promotes conditions suitable for nitrogen-fixing cyanobacteria. Hidrobiologia, 806:411–426, https://doi.org/10.1007/s10750-017-3376-z

Andersen, M. R., K. Sand-Jensen, R. Iestyn Woolway & I. D. Jones, 2017. Profound daily vertical stratification and mixing in a small, shallow, wind-exposed lake with submerged macrophytes. Aquatic Sciences 79: 395–406.

William R. Herb & Heinz G. Stefan (2004) Temperature Stratification nand Mixing Dynamics in a Shallow Lake With Submersed Macrophytes, Lake and Reservoir Management, 20:4, 296-308, DOI: 10.1080/07438140409354159

---

## Author Response (AR2)

Dear authors,

Thank you for your revisions in light of the initial review process. The manuscript and associated documents have been sent back out to the reviewers and they have identified changes and corrections to make before the manuscript is accepted for final publication. Before the manuscript is accepted for final publication, please could you address the comments below and show how these changes have been made throughout the manuscript.

Further, in light of the Reviewer comments, it is identified that the presentation of results should be clarified to ensure that this is clear to non-K-SOM experts. This could be aided by using more explicit definitions within the manuscript text. We added some new information related to K-SOM description. See them in the results section. The preceding discussions could be clarified to address wider related questions (e.g. how can standard Epot estimation procedure be modified to account for changes in the relation between measured air temperature and the temperature of the evaporating surface). The water temperature part was completed by the request of the reviewer. The water temperature was only used to discuss the evaporation difference between pan treatments. As the water temperature is not an easily available meteorological variable, it was not used in analysing the impacting meteorological elements on evaporation as a physical process. This might be a useful next step to take, so some discussion of this would be useful to the readership of HESS. But it was excluded in the aim of the study. The goal was to present the difference in evaporation between standard A pan and planted pans.

Kind regards,

Dr. Daniel Green

Guest Editor, HESS

###

**Reviewer 1 comments:**

General comments:

I appreciate the authors' responses to my suggestions and their work on revising the paper. I still believe that some work on explaining the figures in more plain language would benefit the paper and increase readership. To this end, I would suggest that the authors' explain how to compare the different panels of K-SOM neighbourhood maps in figure 6. They say in their response to review that "These are not traditional Fig. with x and y axis. The Fig. 3 is a schematic layout of correlations, the winning nodes, and their neighbourhood." OK, that is fine, but if I am still confused as a reader, how can the authors improve the communication of the result? We added some extra information related to K-SOM when analysing the heat map, see section 3.2. Perhaps give an example to explain "winning" and "neighbourhood"? There is in section 2.4, first paragraph, lines 3-5. Can the authors explain how to compare the patterns in figure 6? Is it, for example, that if the blue in one hexagon in one figure corresponds to a red hexagon in another figure, then the variables are anti-correlated? A concrete example would be very useful, in my opinion. The following sentence was added to results: "When one variable is red, while the other one is blue on the same place of the heat map, the correlation between them will be negative."

**Reviewer 2 comments:**

General comments:

The revisions have improved the paper, and it's good to see the addition of Figure 5, with the water temperature data compared at clear-sky and cloudy days. However, the data is not used optimally yet in the subsequent discussion. Standard Penman-Monteith equations use meteo-station air temperature, not because it is physically the most relevant parameter (one wants to be closer to the temperature of the surface at which evaporation happens), but rather because air temperatures are commonly measured and plant surface or soil surface temperatures are not. With the water surface temperatures in hand here, authors might propose physically based corrections to standard models (such as Penman-Monteith). As the SD of Ep data was very high in case of P-M method, see also $R^2$ values in Fig. 7, this assumption in Ep estimation is not recommended. Due to submerged macrophytes presence, we chose another standard model, the A pan planted with these crops. These plants are below the water surfaces and not above them. What was questionable in the study, was the increased water loss of evaporation pans due to submerged macrophyte present. This question was answered in the manuscript.

That the K-SOM fitting procedure can account for a much larger share of the day-to-day variability is interesting -- but not very useful for readers of the paper, as no resulting equations (or multi-step algorithms) are provided -- and even if they would, there is no way to know how relevant these would be at another location. If you could show that using the evaporating surface temperature, instead of air temperature, you could close the gap, that would be 'generic'. The relationship between estimated and observed Ep was presented in Fig. 7. The equations were also completed in the Fig. 7. Since the Ep of one sample place was included in the study, the 'generic' impact of submerged macrophytes on Ep was not fully discussed; maybe for different reasons, our results in other sites became variable. More surveys are needed to reveal the applicability of planted standard A pan Ep for different geographical and climatic conditions.

The robust findings that submerged aquatic plants can lead to increased evaporation at a water surface has some direct relevance in limnology; the physical explanation that these effects may act through modified temperature profiles is scientifically satisfying, but there might be a stronger answer to the 'so what?' question if specific adjustments to Penman-Monteith would be proposed. No due to submerged macrophytes below the water surface.

However, despite the preceding discussion, such expectation may not be feasible for the authors -- the minimum requirement for publishing this paper might well be that the basic data are made accessible to others who may be able to take the next steps that will have direct utility to the HESS readership. The data are available for request at the authors.

**Details:**
Title is not yet very attractive:

"Submerged waterplants increase pan evaporation"

"Pan evaporation is increased by submerged waterplants" Done

**Specific changes:**

Line 11 and 64 sipctatum ==> spicatum (as correctly used in line 110) Done

**Reviewer 3 comments:**

The authors provide a revised version of their manuscript taking my comments from the previous review into account. Most of my comments have been addressed, but some comments, especially on the visual interpretation of the K-SOM have not been sufficiently

answered. Additionally, the motivation to use the FAO56-PM is unclear. The motivation of the manuscript was to take into account the impact of the living organism embedded in the standard A pan, in tap water. It is well known that the potential evapotranspiration for plants (not inside the water bodies) can be calculated with the world-wide used P-M equation. In this study, the measured water loss of standard A pans with macrophytes below the water surface was analysed together with calculated potential evapotranspiration by P-M method. As these macrophytes are below the water surface, it was unknown which method may also be closer to the embedded pan evaporation. The accuracy of standard A pan based evaporation was assumed to be closer to the actual water loss of water bodies containing submerged macrophytes than the pot. evapotranspiration of P-M.

It is well known that several methods in evaporation estimation exist. We don't feel the necessity to complete our estimations (measurements and P-M method) with another one. The aim was to study the impact of submerged macrophytes on evaporation and not the calculation of lake evaporation. water loss I suggest the authors to add another method that has been developed explicitly to represent lake evaporation and not reference crop evaporation (see also comment below).

That would be another manuscript dealing with lake evaporation. The P-M determines the potential crop evapotranspiration and not lake evaporation. In case of empty A pans, the evaporation is assumed to be potential. We tried to answer the question – and not more - that how the sediment and submerged macrophytes might change the standard A pan Ep. Additionally, the use of language in the manuscript is poor with frequent incomplete sentences or poorly structured sentences that make the manuscript hard to understand. I provide examples of these below. Thank you, we corrected them. For these reasons, I recommend major revisions before the publication of the manuscript. I provide further comments below.

Additionally, the use of language in the manuscript is poor with frequent incomplete sentences or poorly structured sentences that make the manuscript hard to understand. I provide examples of these below. For these reasons, I recommend major revisions before the publication of the manuscript. I provide further comments below. The examples were taken into account.

**Methods:**

L. 144: "FAO56-PM may also be proper method to get pan evaporation with submerged macrophytes." This is an assumption that may not be true. As the manuscript states it may be proper and not more. And nobody revealed anything about water loss of submerged macrophytes until now. As the authors correctly state, it is a method to estimate crop evaporation, which is a very different application than lake evaporation. As the authors state, the equation they applied was developed for short reference crops (l. 151) which is not the setting here – yes, it is true but this is the standard formula to calculate reference crop evapotranspiration advised by the FAO and widely applied in wetlands (Allen et al. 1998; 2005). After all, this might explain why the PM method yields so poor results and might be simply not suitable to compare evaporation values for the experimental setups in this study. Previously, this fact has not been justified! We wanted to know which estimation is closer to water loss of those water bodies containing submerged macrophytes. Because evaporation physically means the loss of non-living water. There is no question in case of other plants above the water surface; their water loss is (evapo)transpiration and P-M use is accurate. But what is the happening with submerged plants? The FAO-56 method was the closest

method to standard A pan evaporation measurement, as the FAO also suggested the replacement of A pan evaporation by P-M.But not in this case. The authors should instead use a different method that is actually designed to represent lake evaporation. This was not the aim. Even, the effect of the submerged macrophytes is also missing from these formulas. Plenty of literature exists on this topic, see for example Harbeck et al. (1962) and also some others, based on Dalton formula. This new assumption should require a new manuscript with new aim, new processing and methodology. We should refrain from doing so. The authors should also stress that methods like FAO56-PM and Harbeck are special cases of the multiple stepwise regression methods. This latter sentence was taken into consideration (Methodology, 2.3 Section, first paragraph).

L. 183: Why do the authors stress that K-SOM is an indirect method? They state that it is indirect because it estimates evaporation from other quantities. The word "indirect" was emitted from the sentence. The same is true for the other methods so there is no reason to emphasize it for K-SOM. Although the standard A pan measures directly the evaporation. We refined the K-SOM definition in the methodology.

**Results:**

Figure 5: For the SM pan, the difference between surface temperature and 5 cm water temperature is higher between 8 to 16 hours than the difference between surface temperature and 15 cm water temperature. This is surprising to me and should be commented. The stratification of the water temperature in the manuscript is as described in the literature (Jacobs et al. ,1998). The only problem would have been the numbering of the layers (the closest layer to the pan bottom is the 5 cm, and all the others are above this height). The other numbers represent the height of the layer from the pan bottom. The sensor's height was clarified in the title of Fig. 5. We clarified it.

L. 277ff.: The sentence starting with "The inputs..." is a listing of properties. It should be rephrased using simple language. We can't simplify this description. The method requires this detailed explanation. The cited paragraph has been re-written.

L. 300ff.: To me, this paragraph, which describes Figure 6, is still confusing. I have raised this in the previous review already. The authors replied to my comment and I could follow that somehow, but the authors did not made an effort to provide a better description in the manuscript. For readers, who are not familiar with K-SOM, this paragraph will be hard to understand. really, we tried to improve the explanation of Fig. 6.

**Language:**

The language of the manuscript is poor. Sentences are often incomplete or use subjects that are not clear. This makes the manuscript hard to read and prevents publication. For example:

l. 82ff: "Lake evaporation..." There is an article missing in the beginning. The experimental site is not yet introduced because this sentence belongs to the introduction and so it is not possible for the reader to agree with this statement. The paragraph was re-phased. It belongs to the aim that is why it was placed here. The study site introduction is in M&M later.

l. 89: "region" is not introduced and thus, this sentence is not logic. It relates to climate. We corrected with referring to geographical position of Hungary and study site.

l. 90: "Months included in the study (from June to September)." This is not a correct sentence. It misses a verb. corrected

l. 192ff: The sentence starting with "The impact ..." is missing a verb and it is not clear what the authors mean. Thank you, done

l. 300: It should be SM treatment and not MS treatment. done

**Minor comments:**

l. 352: The conclusions from the previous study should be explicitly mentioned. corrected

l. 408: what is the "R" here, is it correlation or something else? Yes, completed

**References:**

Harbeck, G. E. J. (1962). A Practical Field Technique For Measuring Reservoir Evaporation Utilizing Mass-Transfer Theory. Geological Survey Professional Paper 272-E (Tech. Rep.). Washington It seems not the most actual work using the Dalton formula in estimating lake evaporation

---

## Author Response (AR3)

Dear Editor,

Special thanks for your help in improving the manuscript. As requested, a subset of the data has been uploaded as supplementary material.

Best regards,

Brigitta Simon-Gáspár